# Effectiveness of Flavonoid-Rich Diet in Alleviating Symptoms of Neurodegenerative Diseases

**DOI:** 10.3390/foods13121931

**Published:** 2024-06-19

**Authors:** Aneta Szulc, Karolina Wiśniewska, Magdalena Żabińska, Lidia Gaffke, Maria Szota, Zuzanna Olendzka, Grzegorz Węgrzyn, Karolina Pierzynowska

**Affiliations:** Department of Molecular Biology, Faculty of Biology, University of Gdansk, Wita Stwosza 59, 80-308 Gdansk, Poland; aneta.szulc@phdstud.ug.edu.pl (A.S.); karolina.wisniewska@phdstud.ug.edu.pl (K.W.); magdalena.zabinska@phdstud.ug.edu.pl (M.Ż.); lidia.gaffke@ug.edu.pl (L.G.); maria.szota@efekt-ts.pl (M.S.); z.olendzka.082@studms.ug.edu.pl (Z.O.); karolina.pierzynowska@ug.edu.pl (K.P.)

**Keywords:** soy-rich diet, neurodegenerative diseases, flavonoids, animal models, prevention, treatment

## Abstract

Over the past decades, there has been a significant increase in the burden of neurological diseases, including neurodegenerative disorders, on a global scale. This is linked to a widespread demographic trend in which developed societies are aging, leading to an increased proportion of elderly individuals and, concurrently, an increase in the number of those afflicted, posing one of the main public health challenges for the coming decades. The complex pathomechanisms of neurodegenerative diseases and resulting varied symptoms, which differ depending on the disease, environment, and lifestyle of the patients, make searching for therapies for this group of disorders a formidable challenge. Currently, most neurodegenerative diseases are considered incurable. An important aspect in the fight against and prevention of neurodegenerative diseases may be broadly understood lifestyle choices, and more specifically, what we will focus on in this review, a diet. One proposal that may help in the fight against the spread of neurodegenerative diseases is a diet rich in flavonoids. Flavonoids are compounds widely found in products considered healthy, such as fruits, vegetables, and herbs. Many studies indicated not only the neuroprotective effects of these compounds but also their ability to reverse changes occurring during the progression of diseases such as Alzheimer’s, Parkinson’s and amyotrophic lateral sclerosis. Here, we present the main groups of flavonoids, discussing their characteristics and mechanisms of action. The most widely described mechanisms point to neuroprotective functions due to strong antioxidant and anti-inflammatory effects, accompanied with their ability to penetrate the blood-brain barrier, as well as the ability to inhibit the formation of protein aggregates. The latter feature, together with promoting removal of the aggregates is especially important in neurodegenerative diseases. We discuss a therapeutic potential of selected flavonoids in the fight against neurodegenerative diseases, based on in vitro studies, and their impact when included in the diet of animals (laboratory research) and humans (population studies). Thus, this review summarizes flavonoids’ actions and impacts on neurodegenerative diseases. Therapeutic use of these compounds in the future is potentially possible but depends on overcoming key challenges such as low bioavailability, determining the therapeutic dose, and defining what a flavonoid-rich diet is and determining its potential negative effects. This review also suggests further research directions to address these challenges.

## 1. Introduction

Neurodegenerative diseases are a heterogeneous group of disorders characterized by progressive damage and loss of neurons in different areas of the central or peripheral nervous system (CNS and PNS, respectively) [1]. They are accompanied by pathological, histological, and hematological changes [2,3]. The mechanism of neurodegeneration itself (neuronal loss and accompanying disorders) is due to defects at the molecular level, i.e., pathological protein aggregation, synaptic and neuronal network dysfunction, aberrant proteostasis, cytoskeletal abnormalities, altered energy homeostasis, DNA and RNA defects, inflammation, and neuronal cell death [3] (Figure 1). Moreover, every individual has a risk factor leading to the development of changes that favor the onset of neurodegeneration, and that is age. In the ageing brain, there are various processes facilitating neuronal dysfunctions, like increased oxidative stress, abnormalities in cell metabolism (mitochondria-related energy disturbances), alterations in protein processing leading to the accumulation of protein aggregates, and immune system activation [4]. Abnormalities at the molecular and cellular levels translate into a specific clinical picture, i.e., signs and symptoms characteristic for a given disease, in which cognitive impairment, memory impairment, motor dysfunctions, and speaking disability predominate [3,5]. Hundreds of neurodegenerative diseases are now known, but the highest attention is given to the most common ones: Alzheimer disease (AD), Parkinson disease (PD), prion disease, Amyotrophic lateral sclerosis (ALS), Huntington disease (HD), and spinal muscular atrophy (SMA) [6,7]. 

No effective treatment is currently available for any of these diseases, and only some drugs are registered to help reduce or relieve the symptoms [6]. The main difficulty in developing new therapeutics is not only the inability to inhibit neurodegeneration at different levels, but also the need to cross the blood-brain barrier [6]. An advantage would be the possibility of using compounds able to prevent the development of changes facilitating the initiation of neurodegeneration, which appear with age [4]. All the above-mentioned features are possessed by flavonoids [4,5]. These compounds are being extensively studied for their use in the treatment and prevention of neurodegenerative diseases, some of which are proving to be effective not only in alleviating disorders at the cellular level, but also in the clinical picture. 

In this review, we will focus on discussing the effectiveness of the most intensively studied flavonoids. These compounds can be divided into six main groups, namely flavones, flavonols, flavanones, flavanonols, anthocyanidins and isoflavones [4]. Flavonoids are being studied for their potential benefits in conditions such as chronic inflammation, arthritis, cardiovascular disease, diabetes, cerebral ischemia, hemorrhagic conditions including stroke, and even cancer [8,9,10,11,12]. Because of their neuroprotective functions, they are also considered for therapy or prevention in neurodegenerative diseases [13,14,15,16,17,18].

Despite many promising therapeutic effects of flavonoids, low bioavailability remains a significant challenge. These compounds are abundant in common food, such as potatoes, cucumbers, onions, strawberries, seeds, nuts and grapes. However, achieving a therapeutic effect requires consuming these foods in large quantities. For example, an isoflavone-rich diet requires approximately 100 mg of isoflavones per day, which means consuming 250–500 g of tofu, drinking 1–1.7 L of soy milk, or taking 200–500 g of edamame (green soybeans) daily [19]. This is difficult to achieve through diet alone.

Another problem is that the most common source of flavonoids in food is soy and related soy-derived products, while potential adverse effects of their consumption, especially at relatively high amounts, cannot be ignored [20,21]. Especially, isoflavones included in soy act as phytoestrogens, potentially disrupting hormone levels and affecting endocrine function. High soy intake can also interfere with thyroid function due to goitrogens [22]. Moreover, the high fiber content in soy may cause digestive problems, like bloating and gas [23]. Additionally, soy contains phytates that can hinder the absorption of essential minerals, possibly leading to their deficiencies [24]. For those with soy allergies, consumption of soy-derived products can trigger allergic reactions. It was also suggested that excessive soy intake might impact reproductive health and even influence cancer risk, though the latter effect appears controversial [20,21].

Therefore, it is important to investigate whether consuming a combination of polyphenols can provide a synergistic effect, improving bioavailability and reducing the need to consume large amounts of specific foods, taking into consideration also possible adverse effects. It is crucial to address the information gap regarding the combination of products containing various flavonoids to increase their bioavailability. For instance, fisetin, which exerts neuroprotective effects through the gut-brain axis [25], may benefit from being studied in combination with other gut-brain axis compounds or those with high bioavailability, potentially enhancing its absorption and neuroprotective effects. Solving the bioavailability problem could bring us closer to treating or preventing neurodegenerative diseases. Another research goal should be identifying the optimal forms of consuming these compounds through food. Food processing can enhance the therapeutic value and acceptance of these foods, as seen with fermented soy products showing better therapeutic effects than steamed soybean [26,27]. Although flavonoids have relatively low toxicity, excessive consumption may lead to digestive side effects. Proper processing methods, such as heat treatment, can help mitigate these effects. Additionally, it is important to disseminate knowledge about flavonoid consumption among doctors and patients. There are recurring myths about the impact of isoflavones in soy on male feminization [28] and risk of potential interactions with medications, similar to how grapefruit juice can affect drug efficacy [29,30]. For instance, hesperidin, found in large amounts in citrus fruits, including grapefruits, may interact with medications, which is also worth investigating in the case of this flavonoid and the others.

In this review, we present the latest knowledge on neurodegenerative diseases and the most commonly researched flavonoids with an emphasis on therapeutic effects they may cause, with the limitations observed today. 

## 2. Overview of the Use of Flavonoids in Neurodegenerative Disorders

### 2.1. Alzheimer Disease (AD)

AD is a neurodegenerative disorder with a complex etiology. Neurotransmitter abnormalities, accumulation of short β-amyloid segments and aggregation of the hyperphosphorylated form of the tau protein (pTau) into neurofibrillary tangles (NFTs) underlie the disease [31]. In familial forms of the disease (FAD), the occurrence of the aforementioned disorders is associated with mutations in the gene encoding the β-amyloid precursor, APP, or in genes of the prenisilin (*PSEN1* and *PSEN2* genes), which cut the β-amyloid precursor protein APP. Mutations in the *APOE4* gene, which encodes an apolipoprotein responsible for cholesterol transport within the nervous system, are also considered a genetic risk factor for AD [32]. Protein aggregation is accompanied by increased levels of reactive oxygen species (ROS), increased oxidative stress, decreased neurotransmission, and microglial cell-activated neuroinflammation [33]. The first symptom, usually appearing significantly earlier than other disorders, is dementia. Over time, cognitive decline and memory impairment occur [34]. 

Studies conducted on the use of flavonoids have shown that their regular consumption reduced the risk of dementia and AD by lowering the aggregation of pathological proteins [34,35]. However, the efficacy depended on the type of flavonoid used for the therapy. One of the tested compounds was quercetin showing efficacy in improving cognitive function, learning ability and memory in AD [2]. The effect of quercetin was to inhibit the aggregation of both β-amyloid and NFTs and to decrease the level of phosphorylation of the tau protein [36,37]. Improvements in cognitive functions are associated with the inhibition of acetylcholinesterase, an enzyme that degrades acetylcholine [38]. In addition, this compound stimulates antioxidant enzymes and in itself has the ability to reduce ROS levels, thus lowering the oxidative stress [39]. Quercetin also has an inhibitory effect on excessive activation of the immune system by reducing ROS abundance, decreasing the production of pro-inflammatory cytokines in macrophages, and down-regulating the expression of genes encoding them [36]. 

Another compound with proven efficacy in alleviating AD symptoms is rutin, which is a quercetin glycoside. It was demonstrated that rutin improved both behavioral and cognitive impairment [40]. The action of rutin is to alleviate inflammation by inhibiting FβB (nuclear factor kappa light chain enhancer of activated B cells), and NFβB-dependent pro-inflammatory cytokines, and down-regulating the expression of genes encoding pro-inflammatory cytokines [40]. In addition, this compound lowers ROS levels and reduces lipid peroxidation [41]. Administration of rutin also promotes the elevation of levels of markers of neurotrophic factor activity [42].

Genistein is another flavonoid with a therapeutic potential in AD [43]. Like all flavonoids, genistein has antioxidant and anti-inflammatory properties and stimulates autophagy [44]. However, the most important aspect of its action is the ability to lower the levels of the pTau protein, to reduce Aβ deposits and to inhibit the production of this protein [44,45]. In addition, this compound inhibits the activity of acetylcholinesterase, thus demonstrating neuroprotective effects [46]. All of the above-mentioned properties of genistein contribute to its therapeutic effects. Studies using animal models demonstrated that genistein normalized the behavior of animals and improved memory, learning ability and cognitive functions [44,45,46,47]. Moreover, genistein turned out to be effective in delaying the development of dementia in patients with prodromal AD [48].

More flavonoid compounds with their effects and mechanisms of action in AD are shown in Table 1.

### 2.2. Parkinson Disease (PD)

PD is the second most common progressive neurodegenerative disease [49]. Underlying the disease is the accumulation of so-called Lewy bodies composed of α-synuclein and ubiquitin, and Lewy neurites (LN) [50]. A characteristic feature of PD is degeneration of dopaminergic neurons (50–70% degeneration occurs even before the first symptoms) leading to patient-specific motor slowing, rigidity and resting tremor, associated with a decrease in dopamine (DA) level [51,52]. The pathogenesis of PD also includes oxidative stress, high levels of ROS, mitochondrial dysfunction and excessive activation of inflammatory processes [53]. As with other neurodegenerative disorders, there is also no effective treatment of PD, and only partial control of the symptoms and the course of the disease is currently available [54]. 

Myricitrin is being investigated for its potential use not only in the treatment of neurodegenerative diseases, but also in their prevention [55,56]. Research into the mechanism of action of this flavonoid has shown that it effectively reduces oxidative stress, inhibits apoptosis, and reduces the neuroinflammatory response. It is also effective in inhibiting mutant α-synuclein. In addition, after stimulating the activation of inhibitory neurons of the paraventricular nuclei of the hypothalamus, myricitrin exhibits antiepileptic activity. The aim of myricitrin-based therapy of PD is not only to stabilize the abnormalities characteristic for neurodegenerative diseases, but primarily to raise dopamine by increasing efficiency of its synthesis, using agonists for the DA receptor or inhibiting DA degradation [55,56]. Myricitrin has the ability to slow down DA degradation by inhibiting monoamine oxidases (MAO). MAO is a family of oxidoreductases that deaminate catecholamines. In dopaminergic axons, DA from the extracellular space is re-transported for the reuse or degraded by MAO [57]. Myricetin-treated mice showed a reduction in MAO activity and an increase in DA levels in the striatum. Perhaps this compound could be used in the early stages of PD to inhibit the disease progression or to serve as an adjuvant to L-DOPA in more advanced stages [55].

There are promising results from studies on the use of baicalein in the treatment of PD. This compound is known, among other things, for its antioxidant, anti-inflammatory and neuroprotective properties [58]. Studies using animal models demonstrated that administration of baicalein increases the levels of neurotransmitters, i.e., DA, DOPAC (3,4-dihydroxyphenylacetic acid), and HVA (homovanillic acid) [59]. In addition, baicalein protects nervous system cells from 6-hydroxydopamine (6-OHDA)-, 1-methyl-4-phenylpyridinium (MPP+)-, glutamate-, Aβ-, hydrogen peroxide (H_2_O_2_)-, 1-methyl-4-phenyl-1,2,3,6-tetrahydropyridine (MPTP)-, and methamphetamine-induced neurotoxicity [16]. Since one of the main causes of PD is α-synuclein accumulation, baicalein has also been tested for inhibition/degradation of toxic aggregates. It was found that this compound is effective in both inhibiting the aggregation of abnormal proteins and degrading already formed deposits [60]. Importantly, this flavonoid prevented the loss of dopaminergic neurons, and improved the behavioral disturbances that are part of the clinical picture of PD [61,62]. 

More flavonoid compounds with their effects and mechanisms of action in PD are shown in Table 1.

### 2.3. Amyotrophic Lateral Sclerosis (ALS)

ALS is a neurodegenerative disease of a progressive nature. The main cause of ALS is the loss of motor neurons (responsible for voluntary muscle movements) resulting in progressive paralysis leading to death around 5 years after diagnosis. Protein aggregation is also observed during the course of the disease [4]. Despite intensive research, to date only two therapies have been approved with limited efficacy on both motor function improvement and patient survival [63,64]. 

Several studies have been dedicated to the use of flavonoids in the treatment of ALS. One of the compounds tested was fisetin [2]. This compound is a natural antioxidant with anti-inflammatory and neuroprotective properties. The ability to reduce ROS levels translated into a reduction in oxidative stress and neurotoxicity, and preservation of normal mitochondrial functions [65]. Studies in a mouse model of ALS have shown that oral fisetin supplementation improved motor fusions and increased the number of spinal cord motor neurons. It was also observed that fisetin could reduce disease progression and increase life expectancy [66]. 

Another compound whose efficacy is being investigated in the context of ALS therapy is kaempferol. This compound has the ability to inhibit the aggregation of pathogenic proteins, such as Aβ, tau, and α-synuclein, preventing the activation of microglia, reducing pro-inflammatory factors, and lowering ROS levels, possibly through inhibition of the actions of certain enzymes [67]. Kaempferol appears to be a good candidate for ALS therapy due to its ability to counteract neurotoxicity caused by mutations in the *SOD* gene. Aggregation of toxic proteins in cells leads to mitochondrial damage and excessive ROS production. The action of kaempferol is to inhibit the aggregation of the mutant SOD1 protein, to inhibit ROS production and to stimulate autophagy. In addition, it binds to abnormal proteins, preventing the mutant forms from interacting with each other and stabilizing their structure [68,69].

More flavonoid compounds with their effects and mechanisms of action in ALS are shown in Table 1.

### 2.4. Huntington Disease (HD)

HD, a genetic, progressive, neurodegenerative disease, is caused by a mutation resulting in the expansion of the CAG nucleotide triplet in exon 1 of the *IT15* (*HTT*) gene [70]. This leads to creation of a long stretch of glutamine residues in the amino acid sequence of the huntingtin (HTT) protein, which impairs its proper folding. As a result, mutant huntingtin (mHTT) accumulates in cells, including neurons, as insoluble, difficult-to-remove aggregates that impair cell function [71]. Pathological changes in the nervous system are accompanied by neuroinflammation, neurotoxicity, microglial activation, elevated ROS levels, and oxidative stress, leading to mitochondrial dysfunction and impaired neurotransmission [71].

In recent years, genistein has been of great interest to researchers investigating treatments for diseases with neurodegeneration. Similar to the previously mentioned flavonoids, genistein crosses the blood-brain barrier and exhibits neuroprotective effects at multiple levels [72]. Like other flavonoids, genistein has an oxidative effect, reducing oxidative stress by lowering ROS levels in cells and restoring normal mitochondrial function [73,74]. In addition, this compound prevents neuro-inflammatory effects by inhibiting the acetylcholinesterase (AChE) factor-activated B cells’ (NF-κB) activity [73,75]. In the context of the treatment of HD (and other neurodegenerative diseases), an important aspect is the ability of genistein to stimulate autophagy [76]. Autophagy is a process by which, among other things, compounds that are harmful to the cell (like abnormally folded proteins) are removed. This process is often impaired in diseases with accumulation of various compounds, and thus also in neurodegenerative diseases. Genistein improves the efficiency of autophagy, so that mutated huntingtin aggregates are efficiently removed from the cells [76]. Such multi-level effects of genistein lead to improvements in motor, cognitive, behavioral and memory functions [70,73,76].

Another compound from the flavonoid group with a therapeutic potential in HD is luteolin, a substance belonging to the flavones [77]. In traditional Chinese medicine, herbs, vegetables and fruits rich in lutein were used to treat hypertension, inflammation, and cancer [78]. The use of luteolin in the treatment of neurodegenerative diseases is due, among other things, to its anti-inflammatory properties. This compound reduces levels of pro-inflammatory cytokines, interleukins and inflammatory mediators [79]. In addition, luteolin reduces ROS levels and oxidative stress (in HD generated by, among others, the aggregation of mutant huntingtin) by increasing the levels of enzymes and compounds with antioxidant properties, like superoxide dismutase (SOD), glutathione peroxidase and catalase (CAT) [80]. Additionally, in a mouse model of HD, luteolin was shown to stimulate the Nrf2-HO-1 antioxidant pathway [81]. Studies with a cellular model of HD confirmed that luteolin effectively reduced levels of mHTT aggregates and protected cells from the cytotoxic effects of the mutation in *HTT* [81]. Experiments with the *Drosophila* model confirmed the results obtained in studies with cell lines, additionally indicating a delay in the loss of motile functions in flies treated with luteolin [82]. 

More flavonoid compounds with their effects and mechanisms of action in HD are shown in Table 1.

**Table 1 foods-13-01931-t001:** Selected flavonoids and their actions in neurodegenerative diseases.

Neurodegenerative Disease	Flavonoid	Study Model	Therapeutic Effects	References
Alzheimer’s disease	7,8-DHF	animal	rescues memory deficits neuroprotective effect by ameliorating oxidative stress and mitochondrial dysfunction promoted dendrite branching and synaptogenesis inhibited the loss of hippocampal synapses restored synapse number and synaptic plasticity	[83,84,85]
	Catechin	animal	reduce oxidative stress in peripheral and brain tissue decrease alfa-synuclein levels reduce amyloid-β plaque aggregation and tau phosphorylation	[86]
		human	improvement of cognitive function	[87]
	Genistein	animal	reduction of the production of Aβ reduce the accumulation of Aβ1-25 and Aβ3342 fragments. reduce the neurotoxicity induced by Aβ42 and Aβ accumulation significantly improve the cognition activation of autophagy improve cognitive function protective effect of on cholinergic neurons decrease the hyperphosphorylation of tau protein	[45,46]
		human	reduction of the production of Aβ improve cognitive function reduction of inflammation markers	[48]
	Luteolin	animal	exhibits anti-inflammatory neuroprotective properties ameliorated memory and recognitive impairment reduction amyloid-beta (Aβ) production and promotion Aβ degradation amelioration of spatial learning and memory impairment protective effects against amyloid beta-induced oxidative stress	[88,89,90]
		cellular	restored mitochondrial function, decreased mitochondrial superoxide, and preserved mitochondrial morphology	[91]
	Nobiletin	animal	improved cognitive deficits memory improvement protection against cholinergic innervation of the hippocampus reduction of the amount of Aβ and inhibited formation of Aβ plaques ameliorates learning and memory impairment reduced ROS levels in the hippocampus promoting the survival of nerve cells, and promoting axon growth	[92,93,94,95]
Amyotrophic Lateral Sclerosis	7,8-DHF	animal	improved motor deficits preserved spinal MNs and dendritic spines enhanced spatial memory slowed the advance of ALS improve the motor ability promote mitochondrial biogenesis attenuated motor neurons loss	[96,97,98]
	Fisetin	animal	decreases free radical levels to protect against oxidative stress upregulates the expression of antioxidant factors by activation detergent-insoluble mutant hSOD1 proteins improvement of motor ability	[66]
	Genistein	animal	suppressed the production of pro-inflammatory cytokines improved the viability of motor neurons induced autophagy neuroprotective effects by protection against oxygen singlet-induced cerebral damage	[99,100]
	Kaempferide and kaempferol	animal	reduced SOD1 aggregation antioxidant effects prevent neurotoxicity by pumping autophagy	[101,102]
	Quercetin	cellular	inhibit misfolding and aggregation	[103]
		animal	reduced oxidative stress prevents mitochondria	[104]
Parkinson’s disease	Genistein	animal	reduced oxidative stress improvement of spatial learning and memory protection of dopaminergic neurons improved rotational behavior protection against protein aggregation toxicity preserved motor dysfunction	[58,105,106,107]
	Baicalein	cellular	inhibited the formation of α-synuclein oligomers protected cells against cytotoxicity restore mitochondrial function	[58,108,109]
		animal	prevented the progression of α-synuclein accumulation protected DA neurons against α-synuclein oligomer-induced toxicity inhibited the formation of α-synuclein oligomers prevented abnormal behavior by increasing DA neurons, as well as dopamine and serotonin levels in the striatum inhibited oxidative stress and the astroglial response induces mitochondrial autophagy	[110,111,112,113]
	Hesperetin	animal	improvement in learning and cognitive impairment promotes neuronal survival reduced memory loss decrease astrocyte and microglia activation reduced motor dysfunction	[114,115,116,117]
		cellular	suppression of glial activations in primary microglia and astrocytes protection against ROS	[118,119,120]
	Morin	animal	attenuated behavioral deficits and DA deprivation prevented motor dysfunction reduce neuroinflammation ameliorated motor dysfunction alleviated MPTP-induced astrocyte activation	[118,119,121]
	Nobiletin	animal	improved motor and cognitive deficits inhibition of neuroinflammation partial protection of dopaminergic neurons against neurotoxicity	[92]
Huntington’s disease	7,8-DHF	animal	delay motor deficits associated with HD reverse cognitive deficits provide neuronal protection by improving enkephalin levels and preventing volume loss in the striatum normalize levels of induced and neuronal nitric oxide synthase	[122]
	Anthocyanins	animal	delaying motor dysfunction improvement in motor function reduced cholesterol oxidation products in the cortex gender differences in motor function improvement highlight the significance of considering gender disparities in treatment approaches reduction in CAG repeat instability	[123,124]
	Genistein	cellular	correction of HTT levels leads to the disappearance of aggregates formed by mHTT in HD fibroblasts stimulation of autophagy improvement of HD phenotype reduction of mutated huntingtin levels increased cell viability	[76,125]
		animal	antioxidant and anti-inflammatory properties correction of behavioral abnormalities	[126]
	Hesperidin	animal	improved motor function protective effects on the oxidative system, reducing oxidative damage levels and increasing catalase (CAT) enzyme activity inhibiting microglial activation improved behavioral and cognitive function decreasing lipid peroxidation decreasing nitrite concentration increasing superoxide dismutase increasing glutathione levels attenuates neuroinflammation	[127,128]
	Naringin	animal	preventive effects against 3-nitropropionic acid memory function improvement restoration of glutathione system functioning increased intracellular glutathione concentrations reduction of free radicals modulates histological changes reduces the expression of inflammatory mediators	[129,130]

## 3. General Characteristics and Targets of Flavonoids

Flavonoids are a large subgroup of polyphenols which include over 6000 compounds that can be found in products of natural origin, including fruits, vegetables, barks and herbs (tea, berry fruits, apples, red wine, and orange or orange juice). In this review, we will focus on investigating the effectiveness of the most common flavonoids researched. These compounds can be divided into six main groups called: flavones (e.g., luteolin, apigenin, quercetin), flavonols (e.g., quercetin, fisetin, kaempferol), flavanones (e.g., naringenin, hesperidin), flavanonols (e.g., taxifolin, dihydroxyflavonol), anthocyanidins (e.g., cyanidin, pelargonidin, delphinidin) and isoflavones (e.g., genistein, daidzein, glycitein, biochanin A) (Figure 2). This classification is based on the oxidation state of the central C ring, the ring hydroxylation pattern, and the 3-position substitution. Within each group, a diversity of compounds results from the arrangement of hydroxyl groups in combination with glycosylation or alkylation [4]. Due to their numerous properties, for instance antioxidant function, anti-inflammatory activity, and availability to penetrate the blood-brain barrier, flavonoids are increasingly being noticed in the search for therapy for many diseases. In vitro and in vivo studies have demonstrated that flavonoids can penetrate the blood-brain barrier (BBB) and exert effects within the brain. Studies conducted with rats and pigs have specifically shown that one of the exemplary flavonoids discussed in more detail later in this work, anthocyanin, is capable of crossing the blood-brain-barrier [131]. It has also been shown that quercetin, kaempferol, and isorhamnetin are detected in the brain of rats after ingestion of ginkgo biloba extract [132]. Due to their functions, such as anti-apoptotic effects along the Nrf2, NF-κB and p53 pathways, flavonoids play a crucial role in coordinating cellular responses to various stimuli, ultimately determining the survival or death of neurons. Flavonoids are being explored for the treatment of numerous conditions including inflammations, diabetes, and cancer. Moreover, many flavonoids are currently under investigation for their neuroprotective properties, particularly in the context of neurodegenerative diseases [13,14,15]. The strong anti-inflammatory properties of flavonoids in the brain are a promising aspect of therapy for both the most frequent neuroinflammatory diseases, such as AD or PD [16,17,18], but also other disorders, like cerebral ischemia or hemorrhagic conditions, including stroke. 

Because flavonoids belong to a diverse family of plant polyphenolic molecules, it remains unclear whether consuming individual polyphenols or a combination of them (synergistic effects) provides the greatest health benefits to humans. Moreover, studies have revealed the low bioavailability of flavonoid compounds in humans, presenting a significant challenge in establishing recommended intake doses and, consequently, determining their therapeutic value. This challenge is compounded by the declining quality and nutrient density of modern food sources. Despite their possible toxicity, flavonoids are relatively safe and non-invasive drugs of a natural origin. Excessive consumption of flavonoids leading to adverse effects often seems to be related to the overuse of supplements without consulting a doctor or knowing the proper dosage. This is especially a risk for people taking other medications because flavonoids may interact with them [133]. These seem to be extreme cases out of ignorance because flavonoids offer cognitive health benefits in humans, as long-term consumption through food has been shown to reduce the risk of cognitive decline. Cohort studies indicated that flavonoids also lower the risk of AD, PD, and related dementia. Interestingly, the average daily intake of flavonoids from food varies across different regions of the world, for example 132 mg in the United States, 250–900 mg in Europe, 200–650 mg in Asia, and 1650 mg in the Middle East [134]. 

Greater efficacy is observed with the use of plant extracts in supplementation than with the use of synthetic compounds. The challenge with the bioavailability of flavonoids has led to many pharmacokinetic studies in animals and humans and numerous clinical trials. For example, in the case of genistein, nano- and micro-formulations for encapsulation were reported, including lipid nanoparticles, liposomes, tocotrienol-rich nano-emulsions, polymeric nanoparticles, dextran complexes, chitosan complexes [135] or nanoencapsulation of Cyanidin 3-O-Glucoside [136]. Research is ongoing to improve the pharmacokinetics of compounds synthesized ex planta [137].

The presence of hydroxyl and carbonyl groups in flavonoids enables chelation with biological metal ions and interaction with biomolecules, affecting their pharmacological properties. This structural feature allows for designing metal complexes with enhanced biological properties. Flavonoids, weak acids that deprotonate, strongly favor metal ion coordination, influencing reaction kinetics and thermodynamics [138,139]. While metal complexes may enhance antioxidant properties, it is not a universal rule, and each case requires individual examination. This method of modifying flavonoids has proven successful in the case of quercetin, where it has been shown that this treatment supports radical scavenging activity [140]. Increasing the bioavailability of flavonoids and intensifying their action in the human body is certainly an important issue and worth taking action due to their therapeutic potential and pleiotropic functions. Although the exact mechanisms have not yet been fully elucidated, accumulating data increasingly indicate the ability of flavonoids to interact with neuronal receptors, transcription factors and modulate signaling pathways. These pathways include actions of kinases crucial for neuronal activation and synaptic strengthening. As a result, flavonoids are recognized as promising bioactive compounds capable of influencing various aspects of brain functioning, such as synaptic plasticity, cognitive functions improvement, learning and memory in mammalian species [18,141].

It has been suggested that the neuroprotective effects of natural compounds are related to the potential prevention of cellular damage by reducing cellular oxidative stress and inflammation, but also potentially reversing pathological changes already present in the brain [18,142,143]. However, it appears that the molecular mechanisms of actions of flavonoids are significantly more complex. Mechanisms of biological activities of flavonoids vary from one another; however, there are also some common features, presented schematically in Figure 3. In following subsections, we present discussions on several selected flavonoids whose regulatory properties have been investigated most extensively.

### 3.1. Flavones

#### 3.1.1. Apigenin

As one of the most frequently consumed flavonoids from the flavones subgroup, apigenin (4′,5,7-trihydroxyflavone) is present in the highest amounts in parsley, celery, mint, chamomile, lavender, apples, onions, oranges, lemons, broccoli, and spinach. It has a positive effect on learning, memory, and locomotor activity, as well as revealing anxiolytic properties, weaking depressive behaviors, and improving sensorimotor and motor coordination in animals with cognitive disease and neurobehavioral deficits [144,145,146,147]. It has been found that apigenin reduces the reactivity of microglia in the hippocampus, which is important in the initiation and progression of neurodegenerative diseases, such as AD [148]. Apigenin inhibits the loss of NAD^+^ and activation of SIRT3, thus, appropriate homeostasis and mitochondrial functions are maintained in the event of neurotoxicity [149]. Apigenin also inhibits the release of pro-inflammatory cytokines TNF-α, IL-6 and the pro-inflammatory enzyme iNOS-1 in induced PD. Additionally, it prevents the reduction of mRNA expression of neurotrophic factors BDNF and GDNF, reduces the aggregation of α-synuclein and increases the level of the TH protein, and also enhances the action of the dopamine D2 receptor in rats with ROT damage [150]. Moreover, apigenin reversed changes in the expression of genes encoding TNF-α, IL-1β, IL-6, IL-10 and TGF-β [151]. It also has a neuroprotective effect by increasing the number of living neurons and reducing the release of cytochrome c, which may suggest inhibition of the apoptosis pathway [152]. This flavone affects ERK/CREB/BDNF signaling, which translates into long-term enhancement and memory functions. It also alleviates cholinergic and acetylcholine deficits in the hippocampus, which are important for cholinergic neurotransmission and required for memory functioning. Additionally, it has antidepressant and anxiolytic effects, which may be related to the inhibition of inflammatory markers (TNF-α, IL-6, IL-1β, iNOS and COX-2) and the activation of NF-κB [144]. One should note that most of the results of experiments with apigenin come from studies on mouse and rat disease models [153]. Apigenin, like most other flavonoids, has the ability to penetrate the blood-brain barrier, while, as in the case of other flavonoids, the greatest challenge is to increase the bioavailability of these compounds [154].

#### 3.1.2. Nobiletin

Nobiletin (3′,4′,5,6,7,8-hexamethoxyflavone) is the main component of polymethoxylated flavones found in citrus fruit peels [16]. Specifically, nobiletin was demonstrated to be a promising compound in ameliorating cognitive deficits, memory impairment, and pathological features associated with AD, such as excessive accumulation of Aβ, hyperphosphorylation of tau and oxidative stress in animal models [92]. Nobiletin at a dose of 10–50 mg/kg was administered daily to rats for 7 days before and after the disease-inducing surgery. The impairment of reference and working memory in Aβ-treated rats was improved by nobiletin treatment, demonstrating its effectiveness. Furthermore, nobiletin treatment improved motor and cognitive deficits in a 1-methyl-4-phenyl-1,2,3,6-tetrahydropyridine (MPTP)-induced PD mouse model [92]. The research results should encourage researchers to continue studies on this compound, which could potentially become a drug for the treatment and prevention of neurodegenerative diseases. It is worth continuing to examine the effects of this compound alone and in combination with other currently researched therapies for neurological diseases.

### 3.2. Flavonols

#### 3.2.1. Quercetine

Quercetin (3,3,4,5,7-pentahydroxyflavone) is one of the most common secondary metabolites in the plant kingdom and one of the most frequently studied compounds. The Latin term for this flavonoid means oak forest. It belongs to the class of flavonols and is not synthesized in the human body. Quercetin is commonly found in the form of glycosides, i.e., it is conjugated to sugar residues. The largest amounts of this compound are found in onions, but it is also present in apples, berries, herbs (such as dill), asparagus, seeds and nuts, as well as in some varieties of tea and wine. It is the dominant flavonoid in the diet [155,156]. 

Quercetin has antioxidant, anti-inflammatory, antiproliferative, anticancer, antidiabetic and antiviral properties. This molecule is lipophilic and can easily cross the blood-brain-barrier. It is a powerful antioxidant that has been studied both in vivo and in vitro and is currently used in various pharmaceuticals [156,157]. Quercetin is known to be used to treat cancer, allergic reactions, inflammation, arthritis, and cardiovascular diseases. This flavonol also plays an important role in platelet aggregation and lipid peroxidation and enhances mitochondrial biogenesis [158]. The mechanism of its action in AD may be related to the direct suppression of Aβ formation and the stability of Aβ fibrils [159]. 

The neuroinflammatory process is suppressed by quercetin, which down-regulates pro-inflammatory cytokines, which include iNOS and NF-κB, thus influencing neuronal regeneration. Quercetin reduces lipid peroxidation, and thus, reduces the risk of oxidative damage to neurons, as seen in studies involving a mouse model of AD, in which cognitive functions improved after 3 months of intraperitoneal quercetin administration [160]. The neuroprotective role of quercetin included in an energy drink was demonstrated. Consumption of quercetic included in the Red Bull drink increased mRNA expression levels and protein levels of Nrf2 and heme oxygenase-1 (HO-1), which appeared to protect the brain against neurotoxicity. This flavonoid protected the rat cerebral cortex against oxidative damage by acting as an anti-inflammatory agent by reducing the levels of the pro-inflammatory marker interleukin 1 beta and DNA damage markers [161].

In the case of dementia, where there is a disruption in iron homeostasis in the brain, quercetin has been shown to inhibit protein aggregation and prevent cell overload with iron. It regulated genes related to iron metabolism, removed free radicals and silenced Fenton reactions leading to ferroptosis [162]. As a result, quercetin may also be used to remove oxidative damage to the brain caused by iron oxide nanoparticle therapy (IONP), which may cause adverse effects [163]. Reports have shown that quercetin improved the antioxidant enzymes and anti-inflammatory cytokines, and also improved mitochondrial function by targeting SIRT1 activity through the SIRT1/AMPK/NF-κB, SIRT1/Keap1/Nrf2/HO-1, and SIRT1/PI3K/Akt pathways [164]. It has been observed that quercetin exerted neuroprotective effects by modulating microRNA expression in microglial cells. This demonstrated its ability to regulate inflammation, differentiation and apoptosis in neurological diseases [165]. Such treatment was proven to protect primary neurons from microglial toxicity and neurodegeneration induced by LPS in mouse models of PD [166].

#### 3.2.2. Fisetin

Fisetin (3,3′,4′,7-tetrahydroxyflavone) is found in grapes, apples, onions, strawberries, and cucumbers [65]. Like other flavonoids, it is taken up by brain cells and crosses the blood-brain barrier and supports synaptic functions in the hippocampus [4,65]. It has shown effectiveness in neurodegenerative diseases such as AD, PD, HD and ALS [167]. Fisetin acts to regulate neuroinflammation, mediated by SIRT1, Nrf2/HO-1 and p-JNK/NF-κB, abolishes oxidative stress and neuroinflammation by regulating inflammatory mediators (IL-1β and TNF-α) and related markers with autophagy (Atg-3 and Beclin-1), which was confirmed by biochemical and behavioral studies in a mouse model of neuroinflammation and memory impairment [168], and supports synaptic functions in the hippocampus [167]. The effect of fisetin on the physiology of mitochondria in hippocampal HT22 neuronal cells under oxytotic/ferroptotic stress was examined and it has been shown that it is possible to restore mitochondrial homeostasis at the level of redox regulation, calcium uptake, biogenesis, and respiration modulation [169]. Fisetin significantly abolished LPS-induced increased ROS/oxidative stress and activated phosphorylated N-c-JUN terminal kinase (p-JNK) in the adult mouse hippocampus. The treatment inhibited LPS-induced activation of inflammatory Toll-like receptors (TLR4)/cluster of differentiation 14 (CD14)/phosphonuclear factor kappa (NF-κB) and attenuated other inflammatory mediators (tumor necrosis factor-α (TNF)-α), interleukin-1β (IL1-β) and cyclooxygenase (COX-2). Additionally, fisetin improved hippocampus-dependent synaptic and memory functions in adult mice after LPS treatment [170]. It also reduced the accumulation of p-tau and Aβ and additionally increased the expression of genes responsible for the formation of neprilysin, which additionally removed Aβ NEP in the brains of mice [171]. Interestingly, fisetin may play a role in the function of the gut-brain axis, positively influencing the intestinal microbiome, thereby providing neuroprotection to the brain [25].

#### 3.2.3. Kaempferol

Kaempferol (3,4′,5,7-tetrahydroxyflavone) occurs widely in apples, strawberries, grapes, tomatoes, beans, and potatoes, *Gingko biloba* leaves, propolis and oregano [172]. It has antioxidant, antimicrobial, anticancer, neuroprotective and hepatoprotective properties and additionally reduces lipid oxidation [101]. In a mouse model, kaempferol improved learning performance and memory deterioration in streptozotocin-treated rats. At the molecular level, streptozotocin-induced neurodegeneration resulted in decreased expression of GAD67, reelin, and phosphorylated NMDAR. However, kaempferol treatment alleviated these changes, increasing their levels up to the values measured in the control group [173]. It was determined that the NLRP3 (NLR family, pyrin domain cleavage 3) inflammasome is involved in the progression of PD in patients and various mouse models, however, kaempferol protected mice against LPS-induced neurodegeneration by inhibiting the treatment of the NLRP3 inflammasome. The reason for this was noted to be that kaempferol decreased the expression of CASP1-related genes and interfered with the assembly of NLRP3-PYCARD-CASP1 complexes. In parallel, the ingredient also reduced the secretion of IL1B, which means that it is one of the causes of neurodegenerative pathologies. Mechanistically, kaempferol promoted autophagy in microglia, causing decreased expression of the gene encoding NLRP3, which in turn inactivated the NLRP3 inflammasome. Interestingly, ubiquitination was involved in kaempferol-induced autophagic degradation of NLRP3, suggesting the connection of ubiquitination and autophagy by kaempferol, which was confirmed by in vivo studies [25]. 

### 3.3. Flavanones

#### 3.3.1. Hesperetin

Hesperetin is a hesperidin ((2*S*)-3′,5-dihydroxy-4′-methoxy-7-[α-L-rhamnopyranosyl-(1 → 6)-β-D-glucopyranosyloxy]flavan-4-one) aglycone, naturally occurring in large amounts in oranges, tangerines, lemons, and traditional Chinese medicinal herbs; in fact, this compound is heavily used in Chinese medicine [117,174]. Studies with humans have shown that the maximum concentration of hesperetin in plasma improves after oral consumption of citrus juices, such as grapefruit and orange juice [175]. It has various pharmacological properties, including antioxidant and anti-inflammatory properties. The latter properties enhance neuroprotection by supporting neuronal survival via phosphatidylinositol 3-kinase and protein kinase Akt B (PI3K-Akt), by inhibiting the pro-inflammatory cytokines (IL-1β, TNF-α and IL-6), and by controlling the p38 mitogen-activated protein kinase (MAPK) signaling pathway. Moreover, it can act by increasing levels of antioxidant proteins such as Nrf2, as shown in an LPS-stimulated mouse model [176,177]. Modulation of the signaling pathway, as well as antioxidant and anti-inflammatory effects of flavanones may contribute to the observed improvement in cognitive and motor disorders, which was reported in animal models of AD treated with hesperetin [117]. 

The neuroprotective effect of hesperetin was tested in rat models, where it was found that doses of 10 and 20 mg/kg administered for three weeks showed significant improvement in learning skills and cognitive functions and a reduction in the level of oxidative stress. A similar study was conducted to evaluate effects of this molecule in Aβ-stimulated AD, where it was observed to reduce neuroinflammation, apoptosis and neurodegeneration induced by oxidative stress [18,178]. Other studies have shown that hesperetin significantly protected cells against increased oxidative stress by inhibiting apoptotic cell death, ROS production and improving the expression of SOD and GSH, which probably caused the activation of Keap-1, Nrf2 and HO-1 [179]. These results strongly indicated that hesperetin ameliorated LPS-induced pathological features of neurodegenerative diseases in vivo (in adult mouse brains) and in vitro (in BV2 and HT-22 cells). Hesperetin also reduced LPS-induced neuronal apoptosis by inhibiting the action of phosphorylated N-terminal c-Jun kinases (p-JNK), B-cell lymphoma-associated protein 2 (Bcl-2) X (Bax), and caspase-3, and promoting Bcl-2 protein levels. Additionally, hesperetin enhanced synaptic integrity, cognitive function, and memory processes by enhancing phosphorylated cAMP response element binding protein (p-CREB), PSD-95 protein, and syntaxin. It was suggested that hesperetin provides neuroprotection by regulating the TLR4/NF-κB signaling pathway that protects against the harmful effects of LPS [176]. 

#### 3.3.2. Naringenin

Naringenin ((2*S*)-4′,5,7-trihydroxyflavan-4-one) is abundant in various citrus fruits, tomatoes, bergamots and other fruits. Some studies have highlighted its neuroprotective properties. In male rats, naringenin was found to prevent anxiety-like behavior, orofacial dyskinesia and neurodegeneration induced by respine. Moreover, this inductor increased oxidative stress markers and neuronal degeneration in the striatum, both of which were alleviated by naringenin [180]. In neonatal mice, propofol-induced anesthesia led to neurodegeneration via apoptosis, while naringenin administration prevented neuronal loss. After exposure to propofol in adulthood, the mice showed long-term neuronal deficits, impaired neurocognitive functions and problems with memory and learning, while naringenin helped to mitigate these effects in the treated group [181]. The neuroprotective effect of naringenin was additionally exhibited in cellular and animal models of PD, when paraquat was used as an inducer of oxidative stress. Naringenin treatment in SH-SY5Y cells resulted in increased cell viability, reduced oxidative stress, and increased mitochondrial membrane potential. In a rat model, naringenin treatment showed significant neuroprotection against paraquat-induced behavioral deficits. This flavanone seems to modulate levels of mRNAs derived from genes coding for DRD2, DAT, LRRK2, SNCA, β-catenin, caspase-3, and BDNF genes. It also increased TH protein abundance and modulated its immunoreactivity in the striatum of rats and zebrafish, where oxidative stress was induced by 6-OHDA and the levels of oxidative stress biomarkers such as CAT, GSH, SOD and ROS decreased [182,183,184].

### 3.4. Isoflavones 

#### Genistein

Genistein (4′,5,7-trihydroxyisoflavone), a naturally occurring isoflavone found in legumes, like soybeans, green peas, chickpeas and peanuts, shares the ability of other flavonoids to cross the blood-brain barrier, making it a subject of intense study in the aspect of neurodegenerative diseases [65,66,72,185]. In studies involving fibroblast from patients suffering from HD, it was showed that the addition of genistein led to notable degradation of mHTT within cells [125]. Furthermore, research has unveiled its capacity to mitigate oxidative stress and neuroinflammation in neonatal mice modeling cerebral hypoxia and ischemia. Genistein, by increasing cell viability, weakened the effect of neuronal damage and apoptosis, increasing the prospects of long-term recovery in mice. The mechanisms involve the activation of the Nrf1/HO-1 pathway while simultaneously inhibiting the NF-κB pathway [186]. Notably, genistein has shown promising properties in reducing cognitive deficits and neuronal loss in rats administered with ketamine to induce neurodegeneration. These effects were achieved through the regulation of apoptosis-related genes such as those encoding Bax, Bcl-2, caspace 3, and phosphorylated GSK-3b and Akt [187]. Genistein can inhibit the activity of NF-κB signaling pathway, which can be activated via TLR4 and which is one of the important elements influencing the development of neurodegenerative disease. This is consistent with the findings demonstrating ability of genistein to disrupt the binding of NF-κB to target DNA, thereby counteracting neuroinflammatory effects in AD [188]. Moreover, genistein reduced the production of NO in rat microglia induced by LPS and lowered the level of IRF-1 and STAT1. This isoflavone has antioxidant properties that can eliminate ROS and prevent neuronal necrosis and apoptosis induced by Aβ [46]. 

Genistein structurally resembles estradiol, thus, it acts through estrogen receptors, imitating the action of estrogen and influencing both the endocrine and nervous systems. This activity extends to the regulation of cognitive processes such as learning, concentration and memory. Genistein has demonstrated efficacy in restoring or enhancing memory functions across various animal models and human studies [189]. It was found that genistein exerts anti-inflammatory effects in microglial cells by activating the GPER receptor (one of estrogen receptors). These findings were supported by data from both the BV2 microglial cell line and primary microglial cultures, which sheds light on the contribution of GPER to genistein-mediated anti-inflammatory effects on microglial after LPS treatment [190]. There are more and more reports confirming that the action of isoflavones is at least partially exerted through the GPER signal transduction pathway [191,192,193,194]. Moreover, by acting through this receptor, which are increasingly recognized for their significance in neurodegenerative diseases [195], genistein also proved its neuroprotective effects. Studies also indicated that genistein induces dendritogenesis via Erα and GPER which is a newly described mechanism in neuronal protection and synaptic plasticity [196,197].

### 3.5. Anthocyanins

#### Cyanidin

Cyanidin (3,3′,4′,5,7-pentahydroxyflavylium), a natural flavonoid compound abundant in various plants like cherries, blueberries, blackcurrants, mulberries, black soybeans and honeyberries, exhibits neuroprotective properties against degeneration caused by oxidative damage. Both in vitro experiments and mouse model studies have demonstrated the ability of cyanidin-3-O-glucoside to reduce the mRNA expression of proinflammatory cytokines such as IL-1β, TNF-α and IL-6 within the cerebral cortex. Additionally, the treatment has been associated with increased levels of PPARγ and TREM2, leading to the elimination of accumulated Aβ through increased phagocytosis [198]. In a mouse model, oral administration of cyanidin-3-O-glucoside has been shown to protect the brain and improve cognitive behavior by reducing the level of the amyloid precursor presenilin-1, and β-secretase in the cortical and hippocampal regions. Such treatment has been associated with an up-regulation of autophagy-related markers such as LC3-II, LAMP-1, TFEB, and PPAR-α, along with a down-regulation of SQSTM1/p62, improving the autophagy of Aβ plaques and neurofibrillary tangles [199]. Intragastric administration effectively mitigated the decline in brain glucose uptake, alleviated neuronal damage in the hippocampus and cortex, and decreased the Aβ level in the brain [200]. Thus, this anthocyanin derivative may be a promising neuroprotective agent acting by inhibition of glutamate-induced oxidative stress signal and endoplasmic reticulum stress, while activating ERK/Nrf2 pathways [201]. 

Cyanidin treatment has shown significant attenuation of Aβ-induced PC12 cell death and morphological changes by restoring mitochondrial membrane potential through the increased Bcl-2 protein level. Cyanidin also protected cells against DNA damage by blocking ROS and by inhibiting mitochondrial apoptosis [202]. Cyanidin chloride inhibited the upregulation of proinflammatory cytokines TNF-α and IL-6, and suppressed hyperactivity of microglia, induced by LPS, indicating the anti-inflammatory action in the hippocampus. Moreover, this treatment restored the control of expression of genes encoding glial fibrillary acidic protein (GFAP), brain-derived neurotrophic factor (BDNF), glutamate-aspartate transporter (GLAST), and excitatory amino acid transporter 2 (EAAT2), which are members of the excitatory amino acid transporter family [203]. Finally, cyanidin has shown promising features in mitigating bisphenol A-induced oxidative stress and improving impaired neurogenesis in hippocampal and cortical regions of the brain [204].

### 3.6. Flavononols 

#### Taxifolin

Taxifolin (3,5,7,3,4-pentahydroxy flavanone or dihydroquercetin), occurs in many plants, like Siberian pine, Siberian spruce, oak, cedar, walnut but also apples, onions, capers, tea and coffee. It has been shown to be able to limit the formation of Aβ fibrils, protect neurons, maintain normal dopamine levels, improve cerebral blood flow and cognitive abilities. Currently, the health supplement is used in clinical practice. Taxifolin improves cognitive and motor impairment in neurodegenerative diseases such as PD, AD, and HD [205]. The mechanisms underlying taxifolin-mediated neuroprotection against neurodegeneration are presented in Table 1. 

It was demonstrated that oral administration of taxifolin to a mouse model of cerebral amyloid angiopathy (CAA) effectively inhibited the intracerebral production of Aβ by suppressing ApoE-ERK1/2-amyloid-β precursor protein axis, despite its limited permeability through the blood-brain barrier. Taxifolin reduced inflammation by alleviating the accumulation of TREM2-expressing cells in the brain, which are associated with heightened inflammation. I also moderated glutamate levels and attenuated oxidative tissue damage, as well as decreased active caspases, indicating a decrease in apoptotic cell death [206].

## 4. Effects of a Flavonoid-Rich Diet on Neuroprotection

### 4.1. Animal Studies

Experiments employing animal models are especially useful in preclinical studies, when the efficacy of tested compounds can be extensively tested. This is also the case with studies on the effects of a flavonoid-rich diet on neuroprotection.

Animal studies conducted with adult Sprague-Dawley rat females after ovariectomy showed that a soy-rich diet reduced programmed cell death following a 90 min transient middle cerebral artery occlusion (tMCAO). Before tMCAO, rats were fed for two weeks with a diet low in isoflavonoids or a soy-rich diet. The soy-rich diet significantly reduced the extent of infarction and apoptosis in the ischemic brain cortex. A decrease in the translocation of the apoptosis-inducing factor was observed, along with a significant reduction in the number of active caspase-3 positive cells and caspase-mediated α-spectrin cleavage. Additionally, unlike estradiol, the soy diet affects the expression of *bcl-xL* in the ischemic cortex, enhancing its neuroprotective action, suggesting that a soy diet may have a positive impact on protecting the brain from stroke-induced damage [207]. Moreover, a soy diet in male N-Marry albino rats could prevent disturbances in the blood-brain barrier, alleviate increased intracranial pressure and vestibular-motor, and alleviate neurological disorders in traumatic brain injury [208].

Other studies conducted with ovariectomized Sprague-Dawley rats fed for 12 weeks with phytoestrogens from soy sprouts or estradiol demonstrated that both estradiol and phytoestrogens can significantly increase the expression of the brain-derived neurotrophic factor (BDNF)-encoding gene and a gene of its primary receptor TrkB (tyrosine kinase receptors B) in the hippocampus. The increased expression of genes coding for proteins involved in the postsynaptic BDNF-TrkB pathway and presynaptic vesicle proteins (synaptophysin, synaptotagmin 1, and synapsin 1) in the hippocampal tissue had a beneficial effect on the development of synaptic plasticity, synaptic transmission, and overall cognitive function development, which might be responsible for the observed effect of improved spatial learning and memory performance in rats [209]. Similar results of improved spatial memory performance and faster learning of rates were observed in young animals subjected to a 7-week supplementation with a flavonoid-rich berry diet. The cognitive function results correlated with elevated levels of pro- and mature brain-derived neurotrophic factor (BDNF) in the hippocampus [210]. Furthermore, the use of phytoestrogens, unlike the use of estradiol, did not cause negative effects on the reproductive system of animals, suggesting that phytoestrogens with potentially neuroprotective action in the brain may be a safer alternative to estrogen replacement therapy, used among others in postmenopausal women to improve cognitive functions [209,211].

It was demonstrated that genistein has a protective effect on neuron degeneration in a mouse model with ApoE-encoding gene deletion, that was fed a high-fat diet. The absence of apolipoprotein E in the brain leads to increased susceptibility to neurodegeneration. A high-fat diet in mice induces oxidative stress, inflammation of neurons, accumulation of Aβ, hyperphosphorylation of tau, and neuronal cell death. Mice whose diet was supplemented with genistein showed alleviated inflammation of the nervous system by reducing the level of oxidative stress, decreasing Aβ levels, and reducing hyperphosphorylation of the tau protein [212]. It was further observed that enriching the high-fat diet of mice with steamed soy and doenjang, a traditional Korean fermented soybean paste, mitigated inflammation and slowed down neurodegenerative processes. In both diets, as in the case of the use of genistein, a reduction in oxidative stress in the hippocampus and cerebral cortex was observed, as well as altered accumulation of Aβ and hyperphosphorylation of tau. Moreover, it was observed that increased levels of isoflavone aglycone may play a role in improving synaptic plasticity and defending against neurodegenerative diseases. Additionally, it was noted that fermented soy products had greater neuroprotective efficacy than steamed soy, which may be related to the presence of bioactive compounds produced during fermentation [26,27]. 

Numerous studies indicated the effectiveness of both oral and injectable administration of flavonoids in preventing or alleviating symptoms of neurodegenerative diseases [213]. In further analyses, we will focus on the potential of oral application of flavonoids for neuroprotective action, as this way of drug administration is more likely to be used in clinical practice if the treatment is approved.

#### 4.1.1. Animal Models of Alzheimer’s Disease (AD)

In aging mouse models, particularly in SAMP8 mice, there is an early decline in learning and memory along with a series of pathophysiological changes in the brain, including increased oxidative stress, inflammation, vascular disorders, accumulation of Aβ, and hyperphosphorylation of tau [214]. 

It has been demonstrated that administering the flavonol fisetin (at the dose of 25 mg/kg) can regulate the levels of proteins involved in synaptic functions and inflammation, thereby affecting key processes correlated with improved cognitive functions [215]. The impact of flavonoids was also tested with the transgenic AD model 5 × FAD (multiple AD-linked mutations in the genes encoding amyloid precursor protein (APP) and presenilin 1 (PS1)). Chronic administration of the flavone 7,8-dihydroxyflavone (7,8-DHF)-activated TrkB signaling pathways in the brain mitigated synaptic loss, reversed synaptic plasticity, reduced Aβ deposition, and slowed spatial memory deficits by mimicking the action of BDNF, whose main functions are to enhance synaptic transmission, facilitate synaptic plasticity, and promote synaptic growth [83]. The flavone apigenin (at the dose of 40 mg/kg) administered in the 2 × FAD mouse model (AD-linked mutations in genes coding for APP and PS1) for 12 weeks alleviated learning and memory disorders associated with AD similarly to how 7,8-DHF alleviated Aβ burdens, inhibited the amyloidogenic process, oxidative stress, and restored the ERK/CREB/BDNF pathway in the cerebral cortex of mice [216]. 

A rat model of sporadic AD, induced by streptozotocin (STZ), was used to test effects of genistein [21]. The AD rats were treated orally with genistein at a dose of 150 mg/kg/day, followed by a series of behavioral, biochemical, and histological tests. In biochemical studies, the genistein-supplemented group showed normalization of APP and β-amyloid levels, including Aβ40 and Aβ42 forms, as well as Tau and pTau levels. An increase in the levels of LC3-II protein, an autophagy marker, and TFEB, a transcription factor responsible for stimulating lysosome biogenesis, was also observed. Histological analyses revealed a reduction in Aβ aggregates. Administering genistein at a dose of 150 mg/kg/day normalized the defects in the rat brain, induced by STZ, and corrected all behavioral parameters, otherwise deteriorated by STZ injection [45].

#### 4.1.2. Animal Models of Parkinson’s Disease (PD)

The most commonly used models for Parkinson’s disease include the use of toxic compounds 6-hydroxydopamine (6-OHDA), 1-methyl-4-phenyl-1,2,3,6-tetrahydropyridine (MPTP), and 1-methyl-4-phenylpyridinium (MPP(+)). Transgenic models of PD have not been widely used in research due to inconsistent phenotypes of mice with the same mutation [217], however, the MitoPark model appears to be promising [218]. 

The flavone baicalein (at the dose of 200 mg/kg), administered to C57BL/6 mice before exposing them to MPTP, mitigated the loss of dopaminergic neurons, reduced behavioral disturbances, and lowered markers of the oxidative stress and astroglia response [219]. Rats subjected to 6-OHDA and treated with 7,8-DHF (at the doses of 12–16 mg/kg) in their drinking water before and after exposure showed significant behavioral improvement and a reduction in the degradation of dopaminergic neurons in the substantia nigra [220]. In a primate model of PD, intracerebroventricular injections of MPP++ were used, and oral administration of 7,8-DHF at a dose of 30 mg/kg was applied. The flavonoid demonstrated neurotrophic action, protecting dopaminergic neurons from apoptosis in the face of induced neurotoxicity. Moreover, long-term use of 7,8-DHF in monkeys at doses significantly higher than previously used in mice (5 mg/kg) and rats did not show toxicity and did not cause abnormal hematological functions, nor did it impair liver, heart, muscle, kidney, and pancreatic functions [221]. Additionally, other flavonoids like chrysin [222] and nobiletin [223] also demonstrated neuroprotective actions in neurodegenerative diseases, including mouse and rat models of both 6-OHDA- and MPTP-induced PD.

#### 4.1.3. Animal Models of Amyotrophic Lateral Sclerosis (ALS)

Despite promising results in the use of flavonoids for other neurodegenerative diseases, very few flavonoids have been tested on animal models of ALS. The transgenic mouse model SOD1-G93A exhibits protein aggregation, loss of motor neurons, axonal denervation, progressive paralysis, and shortened lifespan [224]. Treatment with fisetin provided neuroprotective action, mitigated motor impairment, reduced ROS-mediated damage, regulated redox homeostasis, and increased survival rates compared to the control group [66].

#### 4.1.4. Animal Models of Huntington’s Disease (HD)

For assessing the neuroprotective nature of flavonoids in HD, both chemical induction of striatal neuron degeneration using 3-Nitropropionic acid (3-NP) and genetic mouse models were employed [225]. 

3-NP is an irreversible inhibitor of mitochondrial complex-II that causes transcriptional dysregulation, bioenergetic failure, protein aggregation, and oxidative damage similar to HD pathogenesis. Chrysin, an active flavonoid, demonstrated neuroprotective efficacy against 3-NP-induced oxidative stress, mitochondrial dysfunction, and neurodegeneration. When administered orally at a dose of 50 mg/kg for 14 days, it improved all behavioral functions, markers of oxidative stress and cell death, and increased the survival of striatal neurons in Wistar rats [226]. Similar neuroprotective effects have been observed with two other flavonoids, quercetin (at the dose of 25 mg/kg) [227] and hesperidin (at 100 mg/kg) [213]. Additionally, neuroprotective effects were observed in the transgenic mouse model R6/1, which exhibits an aggressive phenotype and shortened lifespan. Administration of 7,8-dihydroxyflavone (at 5 mg/kg) delayed the development of motor deficits and reversed deficits in the Novel Object Recognition Test. Furthermore, morphological and biochemical analyses demonstrated prevention of loss in the striatal neurons [122]. In another study with the R6/2 mouse model, phenotypically similar to R6/1, fisetin and resveratrol were used. It was found that both compounds had neuroprotective abilities that helped in preserving normal brain functions in HD through activation and maintenance of the Ras-extracellular signal-regulated kinase (ERK) pathway [228]. 

Recent studies with the R6/1 mouse model of HD also indicated that treatment with genistein at a dose of 150 mg/kg/day, initiated after the first HD symptoms appeared (at 16 weeks of age), has a positive impact on the health of the animals [126]. Genistein treatment corrected emerging abnormalities in both cognitive functions and anxiety levels, as observed in numerous behavioral tests. Additionally, motor functions, which are severely impaired in HD, also improved. There was a reduction in mHTT aggregates and levels of the mutant protein in the brain, likely due to their effective degradation through the autophagy process stimulated by genistein. Oxidative stress levels, as well as pro-inflammatory and anti-inflammatory interleukins, normalized, thereby reducing the inflammatory process, which is a secondary pathomechanism of neurodegeneration in HD. Improvements were observed in biochemical parameters such as GPT/ALAT (glutamic pyruvic transaminase, soluble) and GOT1/ASPAT (glutamic-oxaloacetic transaminase 1, soluble) activities, and levels of bilirubin, CK (creatine kinase), TNN (troponin I), and ROS, which deteriorate with the progressive dysfunctions of organs such as the liver, kidneys, and heart in the course of HD. In summary, genistein administration corrected not only behavioral abnormalities but also most of the selected hematological and biochemical parameters in the blood, skeletal muscles, and heart muscle, whose deterioration was observed in diseased mice [126].

Moreover, animal studies reported positive aspects of a soy-enriched diet on other matters. Eliminating soy from the diet of BALB/c mice for two generations resulted in a stronger anaphylactic reaction and degranulation of mast cells after oral sensitization with peanut extract. Introducing soy or isoflavones into the diet significantly suppressed allergic reactions [229]. Introducing soy milk into the diet of Wistar rats for almost the entire lifespan of the animals caused reduction of the loss of neurons in the hippocampus and extended the lifespan of the animals relative to the control group [230].

### 4.2. Population Studies

Studies on the use of flavonoid-rich diet by humans might be more informative than those conducted with animal models from the clinical point of view. However, the limitations are the restrictions in the use of human subjects in experiments, and the inability to conduct all the test which can be performed with animals. Therefore, interpretation of the results of studies with human subjects are usually more difficult and conclusions are less clear and more indirect, as compared to experiments with animals.

Diets rich in flavonoids include those of the Mediterranean and Asian countries, which are abundant in fruits, vegetables, and legumes. In Asian countries, soy products rich in flavonoids are widely consumed. Studies on the consumption of soy protein and isoflavones among adults from Japan, China, Singapore, and Hong Kong indicated that, on average, in Japan, about 6–11 g of soy protein and 25–50 mg of isoflavones are consumed daily. In Singapore and Hong Kong, consumption is lower than in Japan, while in China, there is considerable regional variation. Moreover, ≤10% of the Asian population consumes as much as 25 g of soy protein or 100 mg of isoflavones (expressed as aglycone equivalents) daily [231]. Prospective population studies conducted by the Japan Public Health Center (JPHC) over a period of 9.4 years found no association between total soy product consumption and the risk of dementia-causing disability in Japanese women and men [232]. The Singapore Chinese Health Study found no link between soy or isoflavone consumption and the risk of cognitive impairment [233]. Similar results were obtained among the elderly population of Taiwan, where soy product consumption was not associated with worsening cognitive functions [234].

A comparison of consuming a soy-rich diet (100 mg of isoflavones daily) versus a low-soy diet (0.5 mg of isoflavones daily) for 10 weeks among young healthy adults of both sexes indicated a beneficial effect of the isoflavone-rich diet on the improvement of short-term and long-term memory and a significant improvement in tests of mental flexibility involving the frontal cortex. Additionally, among women, improvements were observed in tests of letter fluency and planning involving the frontal lobes [235]. Furthermore, prospective cohort studies demonstrated that the consumption of citrus fruits, strawberries, and grapes is associated with a lower risk of stroke in Japanese women [236]. A meta-analysis of preclinical and observational studies, which included both post- and pre-menopausal women, as well as men, concluded that soy isoflavones may improve cognitive functions in adults, especially in the memory domain, and did not indicate the occurrence of serious adverse effects [237].

A randomized, controlled clinical trial concerning the impact of soy isoflavones on cognitive functions in older adults with AD was performed. Individuals diagnosed with early-stage AD underwent a six-month therapy with isoflavones at a dose of 100 mg/day. The patients were evaluated for cognitive functions, including visual spatial memory, visual-motor skills, verbal fluency, and speeded dexterity. The study did not reveal an overall improvement in cognitive functions in the isoflavone-treated group compared to the placebo, and global cognitive functions in both groups deteriorated at a similar rate. During the study, researchers observed significant individual variability in plasma concentrations of genistein, daidzein, and equol in the treatment group, despite adherence to therapeutic recommendations. In the subgroup able to efficiently metabolize the soy isoflavone daidzein to equol, correlations were observed between circulating plasma equol levels and improvement in two cognitive speed-dependent parameters, verbal fluency and speeded manual dexterity. However, due to the small number of participants effectively metabolizing isoflavones among the study population, there is a risk of false interpretation of the results. On the other hand, these studies suggested that soy isoflavones might exert a positive, albeit moderate, impact on cognitive functions among individuals capable of metabolizing isoflavones [238]. It was demonstrated that the ability to metabolize isoflavones was significantly influenced by the intestinal microbiome. This was confirmed by the presence of anaerobic, Gram-positive bacteria, capable of transforming daidzein into S-equol, isolated from the feces of humans and animals capable of metabolizing flavonoids [239]. Studies conducted with 152 older adults from Japan, among whom 40% were capable of effectively converting isoflavones to equol in the intestine, showed that among those able to produce equol, a clearly beneficial impact on cognitive functions occurred [240]. 

In 2022, results of another clinical trial (ClinicalTrials.gov NCT01982578) were published, in which 24 subjects, aged 54–75, with prodromal AD (defined as amyloid-positive minimal cognitive impairment) were given daily oral supplementation of genistein at a dose of 120 mg for 12 months [48]. The results of this trial indicated that patients with prodromal AD treated with genistein for one year showed less amyloid-β deposition in the anterior cingulate cortex (which plays a crucial role in some functional features such as cognitive processes and emotions), as determined by flutemetamol uptake in the specified brain region, compared to patients receiving placebo. Furthermore, the impact of genistein supplementation on the progressive loss of cognitive functions was assessed at two time points, 6 and 12 months of supplementation. At the second time point, a statistically significant improvement was observed in the total dichotomized TAVEC and Centil REY Delayed Copy tests among subjects receiving genistein supplementation. In other tests performed, such as the Mini-Mental State Exam (MMSE), Memory Alteration Test (M@T), Clock-Drawing Test, and Barcelona Test-Revised (TBR), a trend towards improvement was observed. However, making solid conclusions on efficacy of genistein in AD treatment would require further clinical studies on a larger sample size over a longer period than 12 months.

In 2024, results of a meta-analysis focused on the impact of dietary supplementation, food, and dietary patterns on PD were published [241]. The analysis included 24 randomized and crossover clinical trials from 1989 to 2022. It was concluded that diet and dietary supplements had a modest but statistically significant impact on QUICK (quantitative insulin sensitivity check index). The results of the systematic review indicated that Mediterranean, low-fat, and ketogenic diets significantly reduced the total Unified Parkinson Disease Rating Scale (UPDRS) score, while low-protein diets significantly alleviated motor symptoms. 

In a clinical trial conducted in Iran, 70 idiopathic PD patients with an average age of approximately 60 years were investigated (Iranian Clinical Trials Registry IRCT20141108019853N4) [242]. The effect of a Mediterranean diet on PD patients was examined and compared with a control group consuming a traditional Iranian diet. The total antioxidant capacity (TAC) in serum, as well as motor and non-motor aspects of the disease were assessed using the UPDRS scale. The Mediterranean diet significantly increased TAC and lowered the UPDRS score. It was suggested that the Mediterranean diet provides some benefits, but further research is necessary.

Prospective studies with the goal of examining whether higher consumption of all flavonoids and their subclasses (flavanones, anthocyanins, flavan-3-ols, flavonols, flavones, and polymers) was associated with a lower risk of developing PD were conducted. The experiment included 120,617 individuals of both sexes, among whom, over 20–22 years of observation, 805 participants developed PD. The study identified five main dietary sources of high flavonoid content (tea, berries, apples, red wine, and orange juice/oranges). Individuals consuming a high amount of flavonoids in their diet had a 40% lower probability of developing PD, particularly among men [243]. 

Although various cellular and animal studies have evaluated the effect of flavonoids on HD models, a lack of human trial has limited their therapeutic application. In recent years, two significant randomized clinical trials investigated the impact of flavonoids on HD. The first one focused on the effects of consuming 1200 mg of epigallocatechin gallate (EGCG) for 12 months (ClinicalTrials.gov identifier: NCT01357681), assessing its potential neuroprotective and antioxidant benefits in slowing disease progression. The second study examined the impact of administration of 80 mg daily resveratrol on the caudate nucleus volume in HD patients (ClinicalTrials.gov identifier: NCT02336633), exploring its anti-inflammatory and antioxidant properties. Both studies aimed to develop therapeutic strategies for HD, utilizing natural substances to potentially alleviate symptoms or slow progression, providing valuable insights for clinical applications. According to the information available on the ClinicalTrials.gov website (last accessed on 16 May 2024), the data from these studies have not yet been published yet.

A diet rich in flavonoids contained in soy has many positive effects while the negative aspects of an excess of flavonoids in the diet cannot be overlooked. Soy in its raw form contains anti-nutritional factors that alter the amount of amino acids, mineral salts and vitamins absorbed [244]. Soy flour contains a trypsin inhibitor, which can induce hypertrophy and pancreatic hyperplasia. Prolonged exposure to elevated levels of the trypsin inhibitor present in soy resulted in pancreatic nodular hyperplasia and germ cell adenoma in rats subjected to low levels of pancreatic carcinogens [244]. To ensure optimal nutritional properties, soy must undergo a heating process to eliminate internal compounds that adversely affect the gastrointestinal tract [245]. To inactivate the trypsin inhibitors, it is necessary to cleave the two disulfide bonds present within them. This can be accomplished through a variety of processing methods, including physical, chemical, and enzymatic treatments, each conducted under specific conditions of temperature and duration [244]. Dietary supplements containing soy extracts are considered possibly safe for use up to six months. However, soy may cause mild gastrointestinal side effects, including constipation, bloating, nausea and diarrhea. The antinutrients present in soy may compromise the gut barrier function, potentially leading to inflammation and digestive issues [23]. Additionally, it can trigger allergic reactions in some individuals, manifesting as respiratory difficulties, itching and rash [109]. Moreover, concerns exist that soy may negatively affect thyroid function and alter thyroid hormone levels [246]. Dietary soy intake has been associated with an increased risk of bladder cancer and dementia. Due to the activity of estrogen-like substances, it is reasonable for women with ER tumors to be able to safely consume soy and possibly other phytoestrogens. Despite studies suggesting a protective role for phytoestrogens, there is some evidence that genistein, the most abundant isoflavone in soy, may stimulate the growth of estrogen receptor (ER+) breast cancer and interfere with the anti-tumor effect of tamoxifen at low levels [247]. By the same token, soy may also be dangerous when consumed in large quantities (usually in medications) during pregnancy and early childhood development (use of modified soy milk), adversely affecting fetal development and infants during the steroid hormone-sensitive period when estrogen exposure may alter the normal arrangement and function of reproductive tissues [248]. Additionally, soy consumption during pregnancy has been shown to alter the epigenome in offspring, potentially leading to health implications [20]. On the other hand, insufficient soy intake may increase the risk of gestational diabetes mellitus, suggesting that adequate soy consumption could play a beneficial role in preventing this condition [249]. All the above-described effects of an increased uptake of flavonoids present in soy indicate that a soy-rich diet should be used carefully with understanding potential benefits and adverse effects. 

The impact of diet and its components on the body is an interesting and important research topic. Diets characterized by high consumption of fruits and vegetables, legumes, grains, dried fruits, and nuts, with low consumption of red meat, low to moderate consumption of dairy, and moderate to high consumption of fish, have a broad beneficial impact on health. The trend of shifting towards a Western diet, which is characterized by high consumption of saturated fats, highly processed foods, regular consumption of red meat, high content of products with added sugars, and low content of vegetables, fruits, and fiber, is concerning and makes it even more important to pay attention to the beneficial effects of a healthy, balanced diet on the body [250,251]. 

## 5. Concluding Remarks

There are numerous reports indicating that a flavonoid-rich diet might be effective in alleviating symptoms of neurodegenerative diseases. Indeed, flavonoids reveal biological activities which can be particularly helpful in both preventing the symptoms of such diseases and improving functions of the brain in already affected organisms. Studies with animal models confirmed that some soy-derived flavonoids can be effective in treatments of various neurodegenerative diseases, like AD, PD, and HD. Experiments with animal and cellular models indicated specific molecular mechanisms of actions of flavonoids which might explain their efficacy in stimulation of autophagy, inhibition of neuroinflammation, and reduction of the oxidative stress, which lead to correcting neurological defects. On the other hand, studies on the effects of these compounds, including in a diet, in human diseases gave less clear and not so straightforward results. A specific problem is the low bioavailability of flavonoids, especially when used as diet supplements, which has been suggested as one of the major limitations in their potential use as drugs for neurodegenerative diseases. In summary, there is some potential of flavonoids included in foods for the treatment of neurodegenerative diseases; however, further investigations are mandatory to precisely estimate their therapeutic potential and to develop optimal conditions for their potential use in the clinical practice.

## Figures and Tables

**Figure 1 foods-13-01931-f001:**
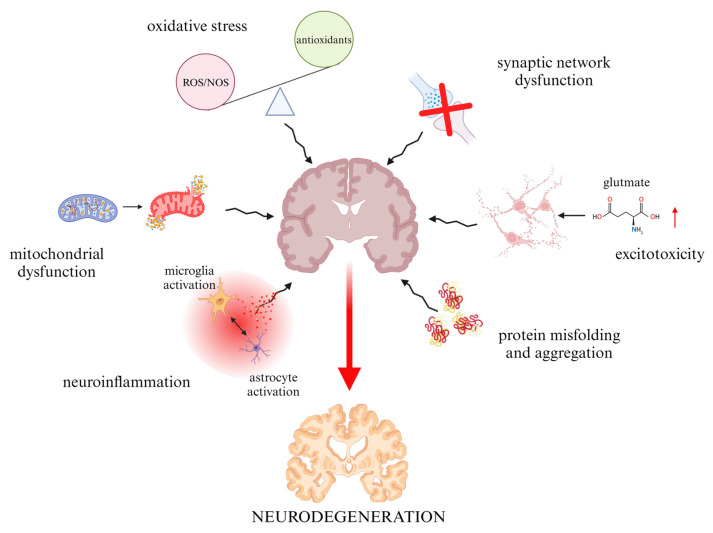
The causes of neurodegeneration. The neurodegeneration process may be caused by various types of abnormalities; the most common of them are (i) increased levels of oxidative stress, (ii) synaptic network dysfunction, (iii) excitotoxity resulting from high glutathione levels, (iv) incorrect folding of proteins and their accumulation in the form of aggregates, (v) neuroinflammation resulting from excessive stimulation of microglia, and (vi) mitochondrial dysfunction causing, among others, increasing concentrations of reactive oxygen species (ROS) and intensifying apoptosis. This figure was made with Biorender.com (accessed on 15 May 2024; Agreement no. YP26TQSS3W).

**Figure 2 foods-13-01931-f002:**
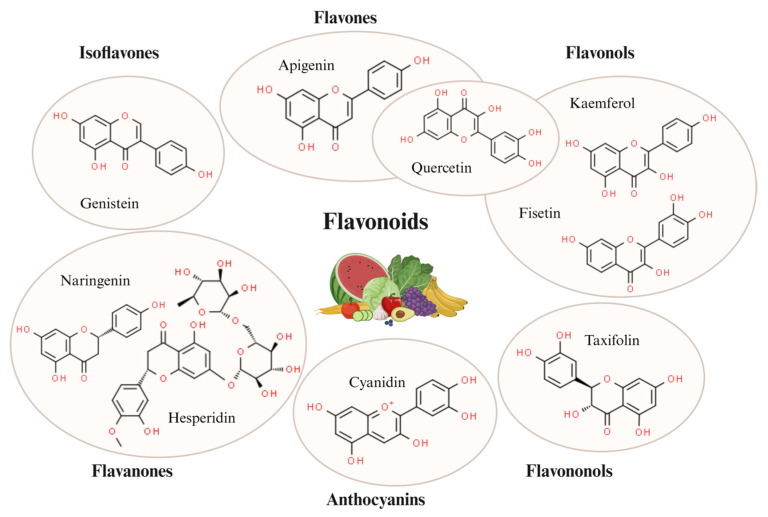
Classification of flavonoids and representatives of each group, described in the text. This figure was made with Biorender.com (accessed on 15 May 2024; Agreement no. GD26TLR8VA).

**Figure 3 foods-13-01931-f003:**
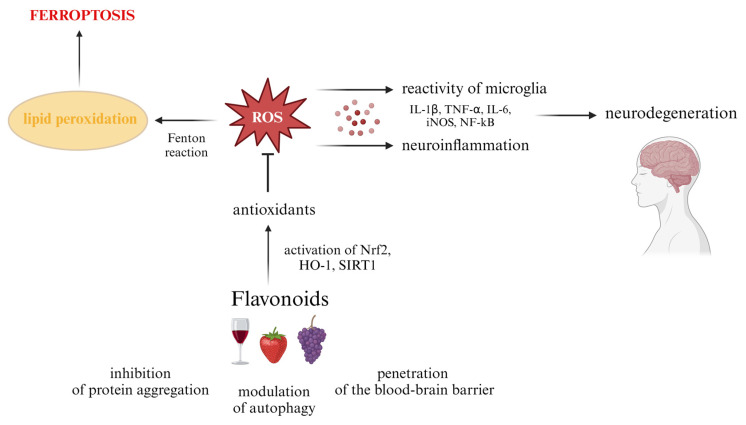
Examples of some of the neuroinflammatory and neuroprotective effects of flavonoids. Flavonoids, due to their ability to modulate autophagy, prevent the formation of protein aggregates and the ability to cross the blood-brain barrier, have important neuroprotective functions, particularly important in neurodegenerative diseases. By activating the Nrf1/HO-1 pathway, as well as improving mitochondrial function by targeting SIRT1 activity, these compounds alleviate the oxidative stress, and effectively reduce ROS formation. When occurring at high levels, ROS can act in various ways, including (i) stimulation of the Fenton reaction, which by increasing lipid peroxidation may contribute to ferroptosis that seems to be important in neurodegenerative diseases, (ii) induction of neuroinflammation which is one of the causes of the initiation and progression of neurodegeneration. Flavonoids inhibit ferroptosis by reduction of ROS levels, and cause a decrease in levels of inflammatory markers (TNF-α, IL-6, IL-1β, iNOS, and NF-κB), positively influencing the survival of neurons and silencing neuroinflammation. This figure was made with Biorender.com (accessed on 15 May 2024; Agreement no. SC26TLT68T).

## Data Availability

No new data were created in this study. Data sharing is not applicable to this article.

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
