# Peer review of "Effectiveness of Flavonoid-Rich Diet in Alleviating Symptoms of Neurodegenerative Diseases"

_foods, 2024, doi:10.3390/foods13121931_

Round 1

Reviewer 1 Report

Comments and Suggestions for Authors

The aim of the manuscript entitled “Effectiveness of soy-rich diet in alleviating symptoms of neurodegenerative diseases” is to investigate and discuss the potential effectiveness of a soy-rich diet, specifically focusing on the flavonoids present in soy, in preventing and alleviating the symptoms of neurodegenerative diseases such as Alzheimer's disease, Parkinson's disease, Huntington's disease, and amyotrophic lateral sclerosis. The study aims to explore the role of dietary flavonoids in managing the onset and progression of these diseases, considering the complex pathomechanisms and varied symptoms associated with neurodegenerative disorders.

The manuscript is very interesting and well-written in general. It is in the scope of the journal. This topic is highly important. My specific comments are given below.

The abstract mentions "issues related to neurodegenerative diseases" without specifying what these issues are. It should be added. Also, the title mentions a "soy-rich diet," but the abstract does not clearly establish a strong focus on soy specifically. It refers generally to flavonoids and their impacts, which could dilute the central theme.

Line 18: “Since neurodegenerative diseases remain incurable” should be substituted with the following: “Currently, most neurodegenerative diseases are considered incurable.”

The abstract does not highlight any new findings or unique perspectives the manuscript might offer, implying a lack of novelty. This should be addressed carefully.

An introduction should be added in order to provide context and background, identify the research gap, and state the study's objectives, scope, and novelty.

The main issue is that this manuscript is not focused specifically on soy flavonoids but on flavonoids in general. The authors have two options. One is to change the title to reflect the broader focus on flavonoids rather than soy alone. For example, "Effectiveness of Flavonoid-Rich Diets in Alleviating Symptoms of Neurodegenerative Diseases." Another, more difficult, but also better, solution is to restructure the manuscript, subtitles, and include more specific details on soy-derived compounds. Also, it is important to discuss the potential risks of high soy dietary intake. Consuming too much soy in the diet can lead to several potential issues. Isoflavones in soy act as phytoestrogens, potentially disrupting hormone levels and affecting endocrine function. High soy intake can interfere with thyroid function due to goitrogens, and its high fiber content may cause digestive problems like bloating and gas. Additionally, soy contains phytates that can hinder the absorption of essential minerals, possibly leading to deficiencies. For those with soy allergies, excessive consumption can trigger allergic reactions. Some studies also suggest that excessive soy intake might impact reproductive health and, controversially, influence cancer risk. Therefore, it is important to discuss the abovementioned points.

Comments on the Quality of English Language

Minor changes are required. 

Author Response

REVIEWER’S COMMENT

The abstract mentions "issues related to neurodegenerative diseases" without specifying what these issues are. It should be added. Also, the title mentions a "soy-rich diet," but the abstract does not clearly establish a strong focus on soy specifically. It refers generally to flavonoids and their impacts, which could dilute the central theme.

RESPONSE:

The tile and abstract have been modified, as recommended by the reviewer.

REVIEWER’S COMMENT

Line 18: “Since neurodegenerative diseases remain incurable” should be substituted with the following: “Currently, most neurodegenerative diseases are considered incurable.”

RESPONSE:

The sentence has been modified as requested by the reviewer (line 19 in the revised manuscript).

REVIEWER’S COMMENT

The abstract does not highlight any new findings or unique perspectives the manuscript might offer, implying a lack of novelty. This should be addressed carefully.

RESPONSE

The abstract has been modified extensively, as recommended by the reviewer (see lines 19-40).

REVIEWER’S COMMENT

An introduction should be added in order to provide context and background, identify the research gap, and state the study's objectives, scope, and novelty.

RESPONSE

As recommended by the reviewer, the chapter “Introduction” has been added.

REVIEWER’S COMMENT

The main issue is that this manuscript is not focused specifically on soy flavonoids but on flavonoids in general. The authors have two options. One is to change the title to reflect the broader focus on flavonoids rather than soy alone. For example, "Effectiveness of Flavonoid-Rich Diets in Alleviating Symptoms of Neurodegenerative Diseases."

RESPONSE

The title has been modified exactly as suggested by the reviewer.

REVIEWER’S COMMENT

Another, more difficult, but also better, solution is to restructure the manuscript, subtitles, and include more specific details on soy-derived compounds. Also, it is important to discuss the potential risks of high soy dietary intake. Consuming too much soy in the diet can lead to several potential issues. Isoflavones in soy act as phytoestrogens, potentially disrupting hormone levels and affecting endocrine function. High soy intake can interfere with thyroid function due to goitrogens, and its high fiber content may cause digestive problems like bloating and gas. Additionally, soy contains phytates that can hinder the absorption of essential minerals, possibly leading to deficiencies. For those with soy allergies, excessive consumption can trigger allergic reactions. Some studies also suggest that excessive soy intake might impact reproductive health and, controversially, influence cancer risk. Therefore, it is important to discuss the abovementioned points.

RESPONSE

Although we have followed the first alternative proposed by the reviewer, focusing on flavonoid-rich diet rather than soy-rich diet, we have also partially followed the latter suggestion. Especially, as soy is the food product richest in flavonoids, we have discussed shortly possible side effects of a soy-rich diet, in fact, using the arguments provided by the reviewer (we thank the reviewer for this very much). See lines 99-109 and 975-1008.

Reviewer 2 Report

Comments and Suggestions for Authors

“Effectiveness of soy-rich diet in alleviating symptoms of neuro-degenerative diseases.”

This article provides an assessment of animal models of neurodegenerative diseases including AD/HD/PD/ALS and considers the properties of a range of flavonoids and the effects that these have been observed to exert on these models. The article then discusses the human benefits of a soy-rich diet or soy isoflavone supplementation, focussing specifically on the flavonoid genistein. The article is detailed and well written, but the broad content (not reflective of the title) and the complex structure impede narrative flow. Some comments/suggestions relating to each section of the review are presented below.

Review:

Title:

Currently the title does not truly reflect the content of the review. Either the title needs to be broader to encompass the current scope of the review, requiring other flavonoid-rich diets to also be considered in the human section, or the review could be limited to soy throughout.

Abstract

Currently the abstract contains no mention of the conclusions nor specific flavonoids that show promise. An abstract should seek to present an overview of the structure and conclusions drawn from the literature presented. However, this abstract appears to be more of a summary of the first section of the review only. From this abstract it appears that flavonoids are a secondary focus to neurodegenerative diseases (NDs) and there is no mention of the purpose of review.

1.       Overview of the use of flavonoids in neurodegenerative disorders

This introduction is well laid out with a good background section, however more focus is needed on why this review is necessary and what the aim of this review is. The section presenting the effects of flavonoids on NDs is good, however the specific papers cited and explained in terms of flavonoids effects on NDs could be moved into the section on human studies. This would provide section 4 with more substance than there is currently. Trying to cover all of these NDs is adding a lot of density to this paper, and trying to include all flavonoids relevant to all of these NDs is not likely to be feasible in one paper alone.

Table 1

From this table it appears that the majority, if not all the flavonoids included are neuroprotective against neurodegenerative diseases, so the subsequent focus on soy-diets needs clearer justification.

2.       General Characteristics of Flavonoids

The first sentence “over 6000 compounds” highlights that the presented review is not fully comprehensive so a justification is needed for only including 10 compounds of interest. Potentially if the paper was set out with a focus “to investigate the efficacy of the most common flavonoids consumed/researched” it would be more feasible to include everything you have and reasonably draw correlates to the relationship between NDs and flavonoid supplementation/dietary intervention.

3.       Target pathways of Flavonoids

Overall in this section, descriptions of mechanisms of actions are good with a comprehensive overview of the genetic regulations as well as neuroinflammatory/redox activity per flavonoid included. However, the introductory section (3. Up to 3.1) could be combined into the general characteristics section 2. above by changing the heading of 3. to reflect the remaining content

Note that other flavones such as luteolin as well as multiple other anthocyanins, isoflavones, flavonols and flavononols are also known neuroprotectives with similar MOAs to e.g., Quercetin and Genistein and others. If there is a reason for excluding these from the literature review then it needs to be explained. For example, if they show no effect on NDs this needs to be explained otherwise comprehensiveness is lacking.

4.       Effect of a flavonoid-rich diet on neuroprotection

4.1 Animal Studies

This section lacks specific information such as a clear definition of a “soy-rich diet”. What does this correlate to in terms of isoflavones concentration such as has been provided in later sections?  Also in 4.1. no other flavonoid sub-group is mentioned until 4.1.1. – this section potentially needs to be renamed into a sub-4.1. header focusing on phytoestrogenic neuroprotection.

4.1.1.-4.1.4.

Overall, these sections are well laid out however without a systematic search and relevant PRISMA flow chart there is no way to confirm the comprehensiveness of the review e.g., 6 animal studies focusing on PD and flavonoids seems low. Similarly, only 6 studies are cited relating to animal models of AD and specific flavonoids. A brief search on scopus using APP and flavonoid has returned 208 studies – if this is the aim of the review then this is in no way comprehensive, as only one genetic strain of mouse used to model AD is included.

4.2 Population studies

This section seems to be the only one to focus on “soy-rich diets” and the main conclusions are seemingly drawn from this (See notes on section 5. below).

Again, there is only one reported PD study in humans looking at the effect of flavonoids – Scopus search of “Parkinsons” “Human” “Flavonoid” has returned 948 studies.

This section seems a bit rushed and not nearly as detailed as the animal sections which have at least have 5 more studies per ND sub header.

5.       Concluding remarks

The  concluding remarks do not appear to fully reflect the subject matter contained within this review. The review focuses on not just soy-rich diets but a plethora of other flavonoids as well which are glossed over in the concluding remarks.

Line beginning 874 comments only on soy benefits but your review has highlighted that other flavonoids are also beneficial.

Line beginning 878: There are hundreds of studies of flavonoids effects on NDs. Whilst they haven’t been included in this review it is false to state “On the other hand, studies on effects of the compounds in human diseases are relatively scarce”.

Line beginning 881: Optimal dose per flavonoid is indeed still open however the question of why bioavailability varies is not – for example comparing an isoflavone to a flavonol is erroneous as they have differing MOAs, differing receptor bindings and binding strengths, differing gut microbiota metabolism, differing rates of absorption and differing rates of peripheral deposition so grouping these all together is not accurate.

Line beginning 885: “Huge potential” is directly contradictory to the statement that isoflavones have a “positive, albeit moderate, impact” (line 840).

Additional notes

Promoting a soy-rich diet but only focussing on Genistein is not fully representative. Daidzein and Glycitein are also present in soy dietary isoflavones as aglycones, glucoside, acetylglucoside and malonylglucoside forms.

The notion of a soy-rich diet is not supported by some of the papers cited. For example, Vina et al (2022) uses a 120mg/day Genistein supplement which cannot be correlated to a “soy-rich” diet.

Author Response

REVIEWER’S COMMENT

Title:

Currently the title does not truly reflect the content of the review. Either the title needs to be broader to encompass the current scope of the review, requiring other flavonoid-rich diets to also be considered in the human section, or the review could be limited to soy throughout.

RESPONSE

As recommended by the reviewer, we have changed the title to reflect the flavonoid-rich diet.

REVIEWER’S COMMENT

Abstract

Currently the abstract contains no mention of the conclusions nor specific flavonoids that show promise. An abstract should seek to present an overview of the structure and conclusions drawn from the literature presented. However, this abstract appears to be more of a summary of the first section of the review only. From this abstract it appears that flavonoids are a secondary focus to neurodegenerative diseases (NDs) and there is no mention of the purpose of review.

RESPONSE

The abstract has been modified extensively, as recommended by the reviewer (see lines 19-40).

REVIEWER’S COMMENT

  1. Overview of the use of flavonoids in neurodegenerative disorders

This introduction is well laid out with a good background section, however more focus is needed on why this review is necessary and what the aim of this review is. The section presenting the effects of flavonoids on NDs is good, however the specific papers cited and explained in terms of flavonoids effects on NDs could be moved into the section on human studies. This would provide section 4 with more substance than there is currently. Trying to cover all of these NDs is adding a lot of density to this paper, and trying to include all flavonoids relevant to all of these NDs is not likely to be feasible in one paper alone.

RESPONSE

We have modified the text according to the reviewer’s suggestion. One major change was to include the new chapter “Introduction” where some parts of the paper were moved to, making the text more clear.

REVIEWER’S COMMENT

Table 1

From this table it appears that the majority, if not all the flavonoids included are neuroprotective against neurodegenerative diseases, so the subsequent focus on soy-diets needs clearer justification.

RESPONSE

We have changed the focus from soy-diets to flavonoids as compounds present in foods. Thus, the text has been modified accordingly.

REVIEWER’S COMMENT

  1. General Characteristics of Flavonoids

The first sentence “over 6000 compounds” highlights that the presented review is not fully comprehensive so a justification is needed for only including 10 compounds of interest. Potentially if the paper was set out with a focus “to investigate the efficacy of the most common flavonoids consumed/researched” it would be more feasible to include everything you have and reasonably draw correlates to the relationship between NDs and flavonoid supplementation/dietary intervention.

RESPONSE

We have addressed this comment by indicating that “In this review, we will focus on discussing the effectiveness of the most intensively studied flavonoids.” (lines 84-85).

REVIEWER’S COMMENT

  1. Target pathways of Flavonoids

Overall in this section, descriptions of mechanisms of actions are good with a comprehensive overview of the genetic regulations as well as neuroinflammatory/redox activity per flavonoid included. However, the introductory section (3. Up to 3.1) could be combined into the general characteristics section 2. above by changing the heading of 3. to reflect the remaining content

RESPONSE:

As requested, formed sections 2 and 3 have been merged, and after including the new section (Introduction), the combined section is a new section 3.

REVIEWER’S COMMENT

Note that other flavones such as luteolin as well as multiple other anthocyanins, isoflavones, flavonols and flavononols are also known neuroprotectives with similar MOAs to e.g., Quercetin and Genistein and others. If there is a reason for excluding these from the literature review then it needs to be explained. For example, if they show no effect on NDs this needs to be explained otherwise comprehensiveness is lacking.

RESPONSE

In fact, in this review article, we have focused on recent studies to avoid redundance with previously published papers. However, we appreciate the reviewer’s comment, and quercetin and genistein are discussed in the light of their neuroprotective activities (lines 154-166, 175-184, 273-287, chapters 3.2.1 and 3.4.1, 735-751, 778-787, 841-865, 921-937).

REVIEWER’S COMMENT

  1. Effect of a flavonoid-rich diet on neuroprotection

4.1 Animal Studies

This section lacks specific information such as a clear definition of a “soy-rich diet”. What does this correlate to in terms of isoflavones concentration such as has been provided in later sections?  Also in 4.1. no other flavonoid sub-group is mentioned until 4.1.1. – this section potentially needs to be renamed into a sub-4.1. header focusing on phytoestrogenic neuroprotection.

RESPONSE

The explanation of the flavonoid-rich diet is present in the newly included chapter “Introduction”. In both Section 4.1, focusing on animal studies, and Section 4.2, describing population studies, we focus first on studies of dietary change and then emphasize animal models of disease or clinical studies of disease. The section names reflect what we wanted to include in a given section.

REVIEWER’S COMMENT

4.1.1.-4.1.4.

Overall, these sections are well laid out however without a systematic search and relevant PRISMA flow chart there is no way to confirm the comprehensiveness of the review e.g., 6 animal studies focusing on PD and flavonoids seems low. Similarly, only 6 studies are cited relating to animal models of AD and specific flavonoids. A brief search on scopus using APP and flavonoid has returned 208 studies – if this is the aim of the review then this is in no way comprehensive, as only one genetic strain of mouse used to model AD is included.

4.2 Population studies

This section seems to be the only one to focus on “soy-rich diets” and the main conclusions are seemingly drawn from this (See notes on section 5. below).

Again, there is only one reported PD study in humans looking at the effect of flavonoids – Scopus search of “Parkinsons” “Human” “Flavonoid” has returned 948 studies.

This section seems a bit rushed and not nearly as detailed as the animal sections which have at least have 5 more studies per ND sub header.

RESPONSE:

We have expanded these sub-sections with additional information. Yes, the Scopus database returns 948 results, but this is a very wide range with a large number of false positive results, and we decided to focus on the latest clinical trials. The latest meta-analysis examining the overall impact of supplementation and diet on PD includes 24 clinical studies from 1989-2022, most of which do not concern a diet rich in flavonoids, which is our main subject of interest. This section is not so rich because there are much fewer clinical studies and they provide less clear results than those conducted on animals, which we mention at the beginning of the section.

REVIEWER’S COMMENT

  1. Concluding remarks

The  concluding remarks do not appear to fully reflect the subject matter contained within this review. The review focuses on not just soy-rich diets but a plethora of other flavonoids as well which are glossed over in the concluding remarks.

Line beginning 874 comments only on soy benefits but your review has highlighted that other flavonoids are also beneficial.

Line beginning 878: There are hundreds of studies of flavonoids effects on NDs. Whilst they haven’t been included in this review it is false to state “On the other hand, studies on effects of the compounds in human diseases are relatively scarce”.

Line beginning 881: Optimal dose per flavonoid is indeed still open however the question of why bioavailability varies is not – for example comparing an isoflavone to a flavonol is erroneous as they have differing MOAs, differing receptor bindings and binding strengths, differing gut microbiota metabolism, differing rates of absorption and differing rates of peripheral deposition so grouping these all together is not accurate.

Line beginning 885: “Huge potential” is directly contradictory to the statement that isoflavones have a “positive, albeit moderate, impact” (line 840).

RESPONSE

We thank the reviewer for indicating these drawbacks.

We have changed “soy-rich” to “flavonoid-rich” (line 1010).

The statement about the scarcity of studies has been modified, to highlight less clear and not so straightforward results of investigations analyzing effects of a flavonoid-rich diet in humans.

The problems of optimal doses of flavonoids and their bioavailability were separated clearly, to avoid any confusion.

“Huge potential” has been replaced by “some potential” to better reflect the results of published studies.

REVIEWER’S COMMENT

Promoting a soy-rich diet but only focussing on Genistein is not fully representative. Daidzein and Glycitein are also present in soy dietary isoflavones as aglycones, glucoside, acetylglucoside and malonylglucoside forms.

RESPONSE

In fact, the literature analysis indicates that the neuroprotective potential of genistein is the highest among isoflavones. However, we have also mentioned other isoflavones in the text.

REVIEWER’S COMMENT

The notion of a soy-rich diet is not supported by some of the papers cited. For example, Vina et al (2022) uses a 120mg/day Genistein supplement which cannot be correlated to a “soy-rich” diet.

RESPONSE

We have modified the focus of this paper to flavonoid-rich diet, thus, a soy-rich diet problems became irrelevant now.

Reviewer 3 Report

Comments and Suggestions for Authors

Based on the title of the manuscript, the authors determined to review the soy-flavonoid compounds in alleviating neurodegenerative diseases. However, there are few points that required deep attention by the authors. 

The abstract does not reflect the title. No mention for soy or any flavonoids compounds extracted from soy. The authors mentioned flavonoids in general without specifying which compounds. 

Line 57, what do the authors mean by ''the clinical picture''? 

The authors have reviewed the pathophysiology for each disease and mentioned a few flavonoids. Table 1 represents all the reviewed studies. However, the authors missed to discuss and include the dosages for each flavonoid which is a critical point. Also, the authors did not include the quantity or the concentrations of these flavonoids in soybeans or in soy extract. The authors mentioned few studies that examine the effect of soy flavonoids in neurodegenerative diseases. But they need to discuss both types of results (negative and positive). 
For example, in line 841, the authors cited a clinical trial where no effects were noticed after six months of consuming soybeans. 

I advise the authors to revise the manuscript and divide it into three sections (clinical trials, in vivo studies, in vitro studies). Also, the authors should mention the flavonoids in soybean in details. 

Author Response

REVIEWER’S COMMENT

The abstract does not reflect the title. No mention for soy or any flavonoids compounds extracted from soy. The authors mentioned flavonoids in general without specifying which compounds. 

RESPONSE
The title and abstract have been modified to reflect the problem indicated by the reviewer.

REVIEWER’S COMMENT

Line 57, what do the authors mean by ''the clinical picture''? 

RESPONSE

This term (the clinical picture) has been explained in the text as “signs and symptoms characteristic for a given disease” which is compatible with the definition of this term (lines 59-61).

REVIEWER’S COMMENT

The authors have reviewed the pathophysiology for each disease and mentioned a few flavonoids. Table 1 represents all the reviewed studies. However, the authors missed to discuss and include the dosages for each flavonoid which is a critical point. Also, the authors did not include the quantity or the concentrations of these flavonoids in soybeans or in soy extract. The authors mentioned few studies that examine the effect of soy flavonoids in neurodegenerative diseases. But they need to discuss both types of results (negative and positive). For example, in line 841, the authors cited a clinical trial where no effects were noticed after six months of consuming soybeans. 

RESPONSE

We would not like to expand the table further. The table is intended to collect the most important information. The doses are discussed in the text, and they vary depending on the model. In the case of animal models, methods of administration were also different (oral or injection).

Regarding the clinical trial mentioned by the reviewer, after 6 months, the authors of the study did not find any significant changes, however, in our opinion, this study was still worth mentioning. We have expanded this description with additional, recent studies, both of which suggested that the number of people participating in the study and the time have a significant impact. Moreover, the observed positive trends need to be assessed in light of subsequent clinical trials, conducted on a larger scale, both in terms of subjects and time.

REVIEWER’S COMMENT

I advise the authors to revise the manuscript and divide it into three sections (clinical trials, in vivo studies, in vitro studies). Also, the authors should mention the flavonoids in soybean in details. 

RESPONSE

We decided to present studies on cells only in the table, paying attention to the potential of flavonoids, but in the context of diet we focus only on animal models with oral administration of flavonoids and on clinical and population studies.

Round 2

Reviewer 1 Report

Comments and Suggestions for Authors

The authors addressed all my comments. 

Comments on the Quality of English Language

Minor changes are required. 

Author Response

This reviewer indicated that the authors addressed all his/her comments. Thus, no specific response is required.

Reviewer 2 Report

Comments and Suggestions for Authors

Overall, this structurally altered version reads better than the previous version. However, not all comments have been fully addressed (see items below).

Abstract

Lines 24-27: “to prevent these diseases altogether” This claim isn’t supported anywhere in the review.

Lines 34-35: “latest knowledge about flavonoids” Again, an overreaching claim given that the presented review is not systematic.

Lines 35-40: “interactions with other compounds” is supported by Lines 126-130 however this is one citation on Hesperidin/Grapefruit that is not mentioned again – if featured in the abstract one would assume there is more evidence than one citation.

Introduction

Lines 86-89: These are all valid therapeutic avenues for flavonoids however these should all be cited individually and not just the final Soy/Cancer citation.

Lines 89-91: This line feels redundant here is it reflects the aim of this review. Could move it to the summary at the end of the introduction section (Lines 131-137).

Lines 103-105: These two statements require citations.

Lines 106-107: Would this not depend on the level of individual allergic reaction? – e.g., some soy-allergic individuals do not require “excessive” consumption to trigger a reaction.

Lines 126-127: The start of this section makes a very good point, the myths of soy intake are of current interest and could potentially be expanded on, incorporating additional facts from the meta-analysis.

Lines 131-133: This needs specifying (see section 3, Lines 310-311) to make it applicable to “the more commonly researched” aim.

Lines 133-137: Proposed avenues of research are not included in detail anywhere in your paper. Simply saying “an avenue of interest” is not the same as recommending future research. The phrase “overcoming the limitations, and thus facilitating” is used but where does the review describe overcoming the bioavailability issues or the interaction issues you mentioned previously?  The closest is in  Lines 358-363 where Genistein encapsulation is described but not reviewed.

2.

Overall this section reads well, is cited well and I am happy with the structural change from version 1.

Lines 185-186 / Lines 229-230 /Lines 260-261 / Lines 303-304 should not read “soy-derived compounds”, there are many other flavonoids included in table 1 and the in preceding sections that these lines are applied to.

Table 1: Again, still happy with this table – I think it summarises the included studies nicely.

3.

Lines 310-311: This alteration is good and describes your included studies well, maybe could do with this around Lines 131-133 in order to set your aim earlier than in the third section.

Lines 340: BioRender requires a citation.

Lines 358-363: This paragraph reports important limitations, namely bioavailability, of flavonoids, but is applicable to supplementation not dietary intake.

Lines 382-383: Not clear why is this intriguing given that it is what you have been explaining for the previous two sections.

Lines 388-390: This is a good specifier for your paper.

Figure 3: “The effects of flavonoids on neuroinflammation and neurodegeneration” needs changing to “An example of some neuroprotective effects” or something similar, as there are other effects – e.g. Genistein’s epigenetic effect of micro-RNA regulations, e.g., miRNA-132 (Kiyama, 2023)

4.

Animal Studies: I found this section informative and well written.

Human Studies:

The review has been renamed and expanded to include all flavonoids, but the flavonoid rich diet section still predominantly reflects soy-rich diets. There is a lot more research on Mediterranean diet and other non-soy flavonoids that could be included here. Perhaps split into separate sections and expand.

Lines 897-900: At what doses?

Lines 910-913: What causes the ability/inability to be an Equol-Producer? Maybe include something about the microbial metabolism just to paint a clearer picture.

Lines 921-953: This is a good inclusion in the paper and provides good information on the structure of these dietary studies, including testing outcomes. However, supplements are mentioned which are outside the purview of a "flavonoid-rich diet" review

Line 935: Spelling mistake - replace "od" with "of"

Lines 975-976: Not sure this is needed. You have previously explained about Asian-diet being rich in soy and Mediterranean diets not. If claiming “the main source”, “at least in some populations” this needs citation or removing.

Lines 994-997: Needs correcting. Current literature describes Genistein’s anti-clastogenic effect in breast cancer (Sharma, 2022) with Daidzein providing an overall protection against clastogenic effects (Snyder, 2003). Citation 252 does not make this claim, in fact they state “Consumption of isoflavones or soy foods is associated with reduced risks of endometrial and bladder cancer” and “It may also reduce the risks of breast and colorectal cancers as well as the incidence of breast cancer recurrence”

Lines 998-1001: Citation needed. Citation 254 does not support this claim.

Concluding Remarks

The conclusions do not align with the abstract or aims. Nowhere does this recommend future avenues of research, nowhere does it overcome limitations and there is very little on any specific flavonoid. The conclusion needs to be restructured, potentially broken down by neurodegenerative disease or flavonoid groups that are beneficial in this endeavour. Overall, this section does not provide a representative summary of your review. You have some good data on the mechanisms of action on some neurodegenerative diseases as well as some efficacious results on dose-dependent studies so why not include this in the summation.

Lines 1019-1023: Again, same comment as version 1, this statement cannot be made regarding all flavonoids grouped together as they all have varying efficacious dosages due to varying mechanisms of action, receptor binding strengths, epigenetic modulation etc. The dosages vary not just between sub-group but within as well, Isoflavones alone have a receptor binding strength difference in relative affinity (compared to 17b-Estradiol) of Genistein (0.49) and Daidzein (0.027) so the dose differences cannot be compared encompassing all flavonoids.

Author Response

Abstract

Lines 24-27: “to prevent these diseases altogether” This claim isn’t supported anywhere in the review.

RESPONSE: According to the reviewer’s notion, this claim has been removed from the Abstract.

Lines 34-35: “latest knowledge about flavonoids” Again, an overreaching claim given that the presented review is not systematic.

RESPONSE: As commented by the reviewer, this claim has been changed. The revised version reads: “this review summarizes flavonoids’ actions and impacts on neurodegenerative diseases” (lines 34-35).

Lines 35-40: “interactions with other compounds” is supported by Lines 126-130 however this is one citation on Hesperidin/Grapefruit that is not mentioned again – if featured in the abstract one would assume there is more evidence than one citation.

RESPONSE: As suggested by the reviewer, this statement has been removed from the Abstract.

Introduction

Lines 86-89: These are all valid therapeutic avenues for flavonoids however these should all be cited individually and not just the final Soy/Cancer citation.

RESPONSE:

As recommended by the reviewer, we have provided more citations.  

Lines 89-91: This line feels redundant here is it reflects the aim of this review. Could move it to the summary at the end of the introduction section (Lines 131-137).

We appreciate this suggestion, however, we modified this sentence rather than moving it at the end of Introduction which should better indicate the neuroprotective functions of isoflavones. The modified sentence reads as follows: “Because of their neuroprotective functions, they are also considered for therapy or preven-tion in neurodegnerative diseases [14–19]” (lines 86-87).

Lines 103-105: These two statements require citations.

RESPONSE:

As recommended by the reviewer, more citations have been added.

Lines 106-107: Would this not depend on the level of individual allergic reaction? – e.g., some soy-allergic individuals do not require “excessive” consumption to trigger a reaction.

RESPONSE:

We agree with this comment, and the text has been modified accordingly. The modified sentence reads as follows: “For those with soy allergies, consumption of soy-derived products can trigger allergic re-actions.” (lines 102-103)

Lines 126-127: The start of this section makes a very good point, the myths of soy intake are of current interest and could potentially be expanded on, incorporating additional facts from the meta-analysis.

RESPONSE:

Although this article is not focused on adverse effects of soy-derived products, the references we cite indicate clearly that myths of the side effects of soy are not necessary true.  

Lines 131-133: This needs specifying (see section 3, Lines 310-311) to make it applicable to “the more commonly researched” aim.

RESPONSE:

As recommended by the reviewer, the text “the most commonly researched” has been added (line 128).

Lines 133-137: Proposed avenues of research are not included in detail anywhere in your paper. Simply saying “an avenue of interest” is not the same as recommending future research. The phrase “overcoming the limitations, and thus facilitating” is used but where does the review describe overcoming the bioavailability issues or the interaction issues you mentioned previously?  The closest is in  Lines 358-363 where Genistein encapsulation is described but not reviewed.

RESPONSE:

We agree with this comment, and the indicated sentence has been removed.  

Overall this section reads well, is cited well and I am happy with the structural change from version 1.

RESPONSE:

We appreciate this comment.

Lines 185-186  / Lines 229-230 /Lines 260-261 / Lines 303-304 should not read “soy-derived compounds”, there are many other flavonoids included in table 1 and the in preceding sections that these lines are applied to.

RESOINSE:

We agree with this comment, and the phrase “soy-derived compounds” has been changed to “flavonoid compounds”

Table 1: Again, still happy with this table – I think it summarises the included studies nicely.

RESPONSE:

We appreciate this comment.

Lines 310-311: This alteration is good and describes your included studies well, maybe could do with this around Lines 131-133 in order to set your aim earlier than in the third section.

RESPONSE:

As recommended by the reviewer, we have mentioned this earlier in the text. The modified text in lines 127-129 reads are follows: “In this review, we present the latest knowledge on neurodegenerative diseases and the most commonly researched flavonoids with an emphasis on therapeutic effects they may cause, with the limitations observed today”

Lines 340: BioRender requires a citation.

RESPONSE:

As requested by the reviewer, we have added agreements’ numbers to legends of all figures prepared with BioRender.com.

Lines 358-363: This paragraph reports important limitations, namely bioavailability, of flavonoids, but is applicable to supplementation not dietary intake.

RESPONSE:

This comment has been addressed by modification of the paragraph. The new paragraph reads as follows: “Greater efficacy is observed with the use of plant extracts in supplementation than with the use of synthetic compounds. The challenge with the bioavailability of flavonoids has led to many pharmacokinetic studies in animals and humans and numerous clinical trials. For example, in the case of genistein, nano- and micro-formulations for encapsula-tion were reported, including lipid nanoparticles, liposomes, tocotrienol-rich nano-emulsions, polymeric nanoparticles, dextran complexes, chitosan complexes [136] or nanoencapsulation of Cyanidin 3-O-Glucoside [137]. Research is ongoing to improve the pharmacokinetics of compounds synthesized ex planta [138].:

Lines 382-383: Not clear why is this intriguing given that it is what you have been explaining for the previous two sections.

RESPONSE:

As suggested by the reviewer, we have removed the indication of “intriguing phenomenon”. The modified text reads as follows: “It has been suggested that the neuroprotective effects of natural compounds are related to the potential prevention of cellular damage by reducing cellular oxidative stress and inflammation, but also potentially reversing pathological changes already present in the brain.”

Lines 388-390: This is a good specifier for your paper.

RESPONSE:

We appreciate this comment.

Figure 3: “The effects of flavonoids on neuroinflammation and neurodegeneration” needs changing to “An example of some neuroprotective effects” or something similar, as there are other effects – e.g. Genistein’s epigenetic effect of micro-RNA regulations, e.g., miRNA-132 (Kiyama, 2023)

RESPONSE:

As recommended by the reviewer, the caption of Figure 3 has been changed, and the current version reads as follows: “Examples of some of the neuroinflammatory and neuroprotective effects of flavonoids.”

Animal Studies: I found this section informative and well written.

RESPONSE:

We appreciate this comment.

Human Studies:

The review has been renamed and expanded to include all flavonoids, but the flavonoid rich diet section still predominantly reflects soy-rich diets. There is a lot more research on Mediterranean diet and other non-soy flavonoids that could be included here. Perhaps split into separate sections and expand.

RESPONSE:

The entire review is extended to include flavonoids, not just soy flavonoids. As can be seen from the table, there are not many studies on the use of individual flavonoids in humans. In this section, we noted that both the Asian and Mediterranean diets are characterized by a large amount of vegetables, fruits, and legumes, which are rich in flavonoids. We described in more detail the studies conducted on Asian populations in terms of the effect on neuroprotection to outline that these reports are not unfounded and because flavonoids are most abundant in legumes. Then, we focused on the neurodegenerative diseases we selected, because the use of flavonoids in these diseases is the main goal of this work. The Mediterranean and Asian diets have many benefits that go beyond the action of the flavonoids present in them, and the beneficial effect of the Mediterranean diet on health is widely described in many works. Nevertheless, we added a short text fragment in the topic of diet as a summary of the chapter and cited works on the Mediterranean diet.

Lines 897-900: At what doses?

RESPONSE:

We did not provide precise doses here, as we discussed meta-analysis of various studies. On the other hand, when citing particular articles, we provided the specific doses.

Lines 910-913: What causes the ability/inability to be an Equol-Producer? Maybe include something about the microbial metabolism just to paint a clearer picture.

RESPONSE:

In response to this question, following text has been added: “It was demonstrated that the ability to metabolize isoflavones was significantly influenced by the intestinal microbiome. This was confirmed by the presence of anaerobic, Gram-positive bacteria, capable of transforming daidzein into S-equol, isolated from the feces of humans and animals capable of metabolizing flavonoids [254].”

Lines 921-953: This is a good inclusion in the paper and provides good information on the structure of these dietary studies, including testing outcomes. However, supplements are mentioned which are outside the purview of a "flavonoid-rich diet" review

RESPONSE:

We did not change this fragment because the cited studies did not specify what kind of genistein was used (probably a plant extract). Earlier in the text, in the fragment about bioavailability, we indicated that supplements made from plant extracts show greater effectiveness and that studies are being conducted on improving the pharmacokinetics of compounds synthesized ex planta (lines 351-358).

Line 935: Spelling mistake - replace "od" with "of"

RESPONSE:

This error has been corrected.

Lines 975-976: Not sure this is needed. You have previously explained about Asian-diet being rich in soy and Mediterranean diets not. If claiming “the main source”, “at least in some populations” this needs citation or removing.

RESPONSE:

This sentence has been removed.

Lines 994-997: Needs correcting. Current literature describes Genistein’s anti-clastogenic effect in breast cancer (Sharma, 2022) with Daidzein providing an overall protection against clastogenic effects (Snyder, 2003). Citation 252 does not make this claim, in fact they state “Consumption of isoflavones or soy foods is associated with reduced risks of endometrial and bladder cancer” and “It may also reduce the risks of breast and colorectal cancers as well as the incidence of breast cancer recurrence”

Lines 998-1001: Citation needed. Citation 254 does not support this claim.

RESPONSE:

As recommended by the reviewer, this section has been corrected. In fact, the whole fragment has been modified (for details, see lines 986-1012).

Concluding Remarks

The conclusions do not align with the abstract or aims. Nowhere does this recommend future avenues of research, nowhere does it overcome limitations and there is very little on any specific flavonoid. The conclusion needs to be restructured, potentially broken down by neurodegenerative disease or flavonoid groups that are beneficial in this endeavour. Overall, this section does not provide a representative summary of your review. You have some good data on the mechanisms of action on some neurodegenerative diseases as well as some efficacious results on dose-dependent studies so why not include this in the summation.

Lines 1019-1023: Again, same comment as version 1, this statement cannot be made regarding all flavonoids grouped together as they all have varying efficacious dosages due to varying mechanisms of action, receptor binding strengths, epigenetic modulation etc. The dosages vary not just between sub-group but within as well, Isoflavones alone have a receptor binding strength difference in relative affinity (compared to 17b-Estradiol) of Genistein (0.49) and Daidzein (0.027) so the dose differences cannot be compared encompassing all flavonoids.

RESPONSE:

The “Conclusion” chapter has been modified, especially in points indicated by the reviewer. However, describing details of the mechanisms, specific flavonoids, and their use in particular diseases, would make this chapter very long, and thus, had to follow. Therefore, while the chapter was modified, we avoided to make it significantly longer.

Reviewer 3 Report

Comments and Suggestions for Authors

The authors have completed the corrections. 

Author Response

This reviewer indicated that the authors have completed the corrections. Therefore, no specific responses are required during this second revision.